Letter

# An unusual tandem kinase fusion protein confers leaf rust resistance in wheat

Yajun Wang [1,2], Michael Abrouk [1,2], Spyridon Gourdoupis [3], Dal-Hoe Koo [4], Miroslava Karafiátová [5], István Molnár [5,6], Kateřina Holušová[5], Jaroslav Doležel [5], Naveenkumar Athiyannan [1,2], Emile Cavalet-Giorsa [1,2], Łukasz Jaremko[3], Jesse Poland [1,2] & Simon G. Krattinger [1,2] ✉

The introgression of chromosome segments from wild relatives is an established strategy to enrich crop germplasm with disease-resistance genes[1]. Here we use mutagenesis and transcriptome sequencing to clone the leaf rust resistance gene *Lr9*, which was introduced into bread wheat from the wild grass species *Aegilops umbellulata*[2]. We established that *Lr9* encodes an unusual tandem kinase fusion protein. Long-read sequencing of a wheat *Lr9* introgression line and the putative *Ae. umbellulata Lr9* donor enabled us to assemble the ~28.4-Mb *Lr9* translocation and to identify the translocation breakpoint. We likewise cloned *Lr58*, which was reportedly introgressed from *Aegilops triuncialis*[3], but has an identical coding sequence compared to *Lr9*. Cytogenetic and haplotype analyses corroborate that the two genes originate from the same translocation event. Our work sheds light on the emerging role of kinase fusion proteins in wheat disease resistance, expanding the repertoire of disease-resistance genes for breeding.

Bread wheat (*Triticum aestivum*; 2*n*=6*x*=42; AABBDD genome) is one of the most widely cultivated crops worldwide[4]. The fungal pathogen *Puccinia triticina* (*Pt*) causes leaf rust, which is among the most widespread and devastating wheat diseases, resulting in annual production losses of ~25 million tons[5,6]. Of the 69 known leaf rust (*Lr*) resistance genes in wheat, 28 were introgressed from the secondary and tertiary gene pools through interspecific hybridizations[7]. The genus *Aegilops* (annual goat grasses) comprises diploid and polyploid species that serve as reservoirs to increase wheat genetic diversity[8,9]. The two resistance genes *Lr9* and *Lr58* were introgressed into bread wheat from the U-genomes of *Aegilops umbellulata* (diploid; 2*n*=2*x*=14; UU genome) and *Aegilops triuncialis* (tetraploid; 2*n*=4*x*=28; U$^t$U$^t$C$^t$C$^t$ genome), respectively[2,3]. The *Lr9* introgression is of historic importance, as it was the first disease-resistance gene transferred into bread wheat in the 1950s using irradiation, although the *Ae. umbellulata* donor was not specified[2]. The resulting bread wheat line, 'Transfer', carried an *Ae. umbellulata* translocation at the end of chromosome arm 6BL[2,10]. The *Lr58* introgression reportedly occurred spontaneously through homoeologous recombination while backcrossing *Ae. triuncialis* accession TA10438 to the susceptible bread wheat line WL711 (ref. 3). *Lr58* was mapped to chromosome arm 2BL in the resulting backcross line TA5605 (ref. 3). Wheat cultivars carrying *Lr9* were released in the late 1960s. *Lr9* is still effective in many wheat growing areas, although virulent *Pt* isolates emerged a few years after the release of the first *Lr9*-carrying wheat cultivars[11]. *Lr58* has not been widely deployed in wheat breeding, and *Pt* races virulent on *Lr58* have been reported[3].

Both *Lr9* and *Lr58* conferred strong resistance against multiple *Pt* isolates in controlled conditions (Fig. 1a and Extended Data Fig. 1). To clone the two genes, we screened ethyl methanesulfonate (EMS)-induced mutant populations in the genetic backgrounds of

[1]Plant Science Program, Biological and Environmental Science and Engineering Division, King Abdullah University of Science and Technology (KAUST), Thuwal, Saudi Arabia. [2]Center for Desert Agriculture, King Abdullah University of Science and Technology (KAUST), Thuwal, Saudi Arabia. [3]Bioscience Program, Biological and Environmental Science and Engineering Division, King Abdullah University of Science and Technology (KAUST), Thuwal, Saudi Arabia. [4]Wheat Genetics Resource Center and Department of Plant Pathology, Kansas State University, Manhattan, KS, USA. [5]Institute of Experimental Botany of the Czech Academy of Sciences, Centre of the Region Haná for Biotechnological and Agricultural Research, Olomouc, Czech Republic. [6]Present address: Agricultural Institute, Centre for Agricultural Research, Martonvásár, Hungary. ✉e-mail: simon.krattinger@kaust.edu.sa

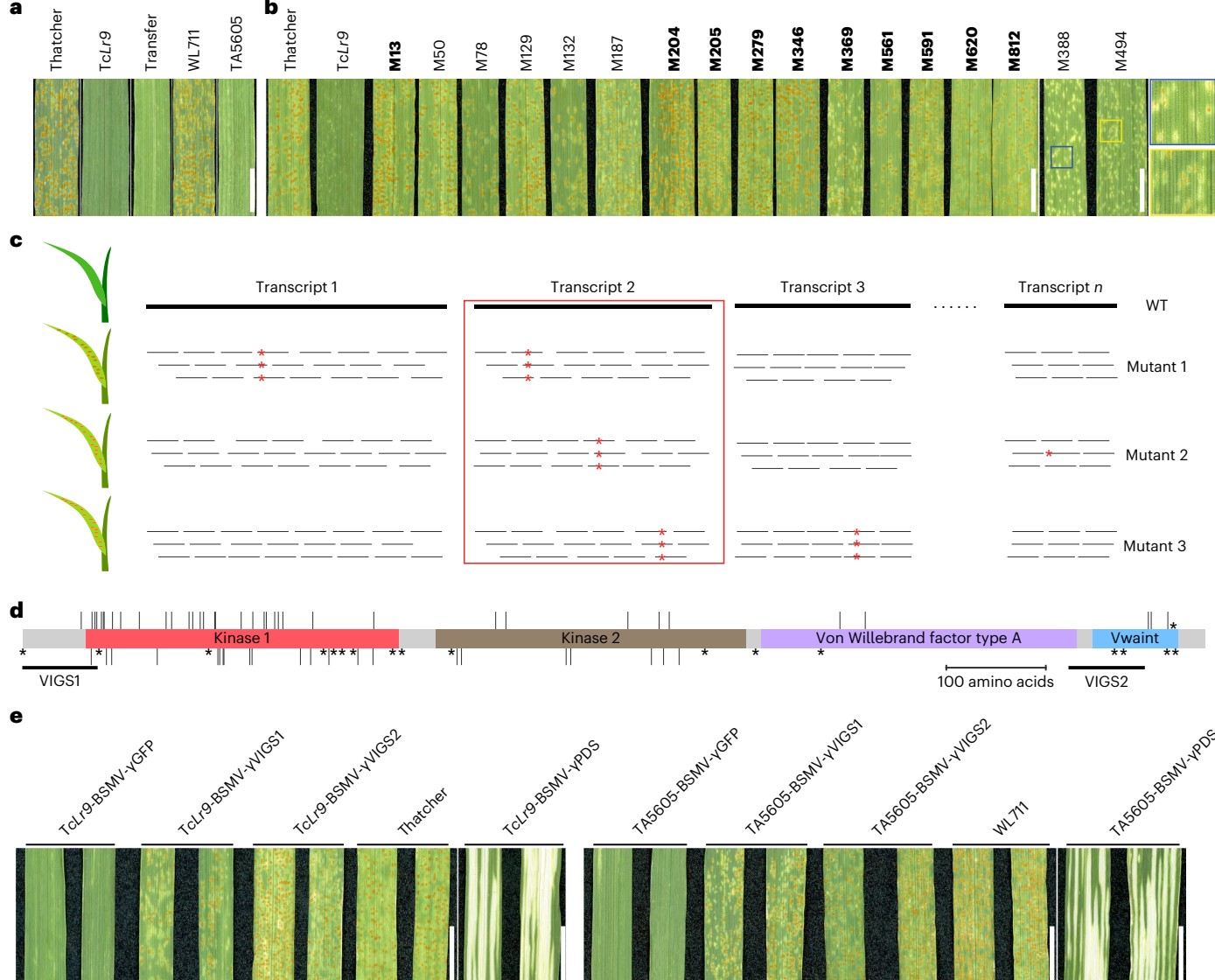

**Fig. 1 | Lr9 and Lr58 encode an unusual tandem kinase fusion protein.**
**a**, Phenotypic effect of *Lr9* and *Lr58* against *P. triticina*. Tc*Lr9* and Transfer represent *Lr9*-carrying bread wheat lines, and Thatcher is the susceptible control. TA5605 is an *Lr58*-carrying bread wheat backcross line in the genetic background of the susceptible parent WL711. Scale bar = 1 cm. **b**, Representative images showing the phenotype of susceptible Tc*Lr9* mutants. Mutants in bold were used for MutIsoSeq. Blue and yellow boxes show magnified sections of the partially susceptible mutants M388 and M494, respectively. Scale bar = 1 cm. **c**, Schematic representation of MutIsoSeq. Iso-seq is used to obtain full-length transcript sequences of a WT parent. RNA sequencing reads generated from EMS mutants are mapped against the WT full-length transcripts and SNPs are called. Transcripts showing an EMS-type point mutation (red stars) in all the mutants are considered candidates (transcript 2 in this example). **d**, The predicted WTK6-vWA protein structure; salmon red−kinase 1, light brown−kinase 2, purple−vWA domain, light blue−Vwaint domain. Vertical bars represent 67 EMS-induced amino acid substitutions, upper bars represent surface-localized residues, lower bars represent internal residues and black stars represent premature stop codon and start codon mutations. VIGS1 and VIGS2 indicate the positions of the two VIGS constructs. **e**, Representative images showing the results of the VIGS experiment using barley stripe mosaic virus (BSMV). BSMV-γVIGS1 and BSMV-γVIGS2 indicate two silencing constructs targeting *WTK6-vWA*, BSMV-γGFP indicates control with a GFP silencing construct and BSMV-γPDS indicates silencing construct targeting the phytoene desaturase genes. Scale bar = 1 cm.

Thatcher*Lr9* (Tc*Lr9*; Transfer/6*Thatcher; 919 M2 families) and TA5605 (*Lr58* carrier; ~7,400 M2 families) for loss of resistance. We identified and validated 17 and 104 susceptible mutants in these two backgrounds, respectively (Fig. 1b and Extended Data Fig. 2). Because alien introgressions often show suppressed recombination[1], we developed MutIsoSeq to clone *Lr9* and *Lr58*. MutIsoSeq combines isoform sequencing (Iso-seq) of WT parental lines and transcriptome deep sequencing (RNA-seq) of EMS mutants to identify altered transcripts for candidate genes and does not require genetic mapping (Fig. 1c). First, we generated Iso-seq reads derived from RNA isolated from *Pt*-inoculated Tc*Lr9* and TA5605 leaves. We then mapped RNA-seq reads from 10 Tc*Lr9* and 10 TA5605 susceptible mutants against their respective parents and identified EMS-type (G/C to A/T) mutations. For both sets of mutants, we only identified one transcript with EMS-type point mutations in all 10 sequenced mutants (Table 1, Supplementary Note 1 and Supplementary Figs. 1 and 2). Surprisingly, the putative *Lr9* and *Lr58* coding sequences were identical (Supplementary Fig. 3a–c) suggesting that (1) two identical genes were introgressed independently from two different U-genome species or (2) *Lr9* and *Lr58* are actually the same original introgression and named differently from pedigree or mapping errors. Several lines of evidence indicate that *Lr9* and *Lr58* are from the same original introgression, that *Ae. triuncialis*

**Table 1 | Number of transcripts showing point mutations across a given number of mutant families**

|  | Tc*Lr9* | TA5605 |
|---|---|---|
| Number of transcripts with no mutation compared to wild type | 177,216 | 124,101 |
| Number of transcripts mutated in one of ten mutants | 9,072 | 18,949 |
| Number of transcripts mutated simultaneously in two of ten mutants | 756 | 2,137 |
| Number of transcripts mutated simultaneously in three of ten mutants | 68 | 214 |
| Number of transcripts mutated simultaneously in four of ten mutants | 2 | 22 |
| Number of transcripts mutated simultaneously in five of ten mutants | 1 | 3 |
| Number of transcripts mutated simultaneously in six of ten mutants | 0 | 0 |
| Number of transcripts mutated simultaneously in seven of ten mutants | 0 | 0 |
| Number of transcripts mutated simultaneously in eight of ten mutants | 0 | 1[a] |
| Number of transcripts mutated simultaneously in nine of ten mutants | 0 | 0 |
| Number of transcripts mutated simultaneously in ten of ten mutants | 1 | 0 |

[a]This transcript was mutated in all ten mutants. However, two of ten mutations were identical (possibly nonindependent mutants). In the MutIsoSeq pipeline, the parameter '-z' (number of mutant lines that are allowed to have a SNP in the same position) was set as 1. Because of this setting, the two identical mutations were not counted and the candidate transcript for TA5605 was detected with eight mutants. Ten susceptible loss-of-resistance mutants in the Tc*Lr9* and TA5605 background were sequenced.

accession TA10438 is not the donor of *Lr58* and that *Lr58* is likely *Lr9* (Supplementary Note 2).

The putative 3,504-bp *Lr9/Lr58* (*Lr9* hereafter) coding sequence encoded a 1,167 amino acid protein with an N-terminal tandem kinase domain followed by a von Willebrand factor A (vWA) domain and a Vwaint domain in the C-terminus (Fig. 1d). Wheat tandem kinases (WTKs) have recently emerged as a new prominent player involved in disease resistance in *Triticeae*, the botanical tribe including wheat and barley[12–18]. The fusion of a WTK to another domain, however, is unique and highly unusual. vWA and Vwaint domains often occur together (subsequently referred to as vWA/Vwaint domain) and are thought to participate in protein–protein interactions in plants[19,20]. The vWA-containing copine proteins in *Arabidopsis* and rice are regulators of disease resistance and the *Arabidopsis* copine protein BON1 might be guarded by the NLR immune receptor SNC1, supporting a potential role of vWA-containing proteins in defense[19,21,22]. The Lr9 kinase domains belong to the LRR_8B subfamily (cysteine-rich receptor-like kinases), which is the most frequent kinase subfamily found in WTK proteins[23]. To reflect the introduced nomenclature, we refer to the *Lr9* candidate as *WTK6-vWA* hereafter. A PCR marker derived from *WTK6-vWA* showed complete linkage with disease resistance in F2 populations derived from crosses between Tc*Lr9* and the susceptible wheat cultivar Avocet S (136 F2 plants) and TA5605 with Avocet S (128 F2 plants) (Extended Data Figs. 3 and 4), confirming that *WTK6-vWA* identified through whole-transcriptome sequencing co-segregates with the gene mediating leaf rust resistance in Tc*Lr9* and TA5605.

We amplified and Sanger-sequenced the *WTK6-vWA* genic region from the remaining 101 *Lr9* mutants and detected missense or nonsense mutations in all but one mutant in the Tc*Lr9* background, which might harbor second-site mutations or a mutation in the regulatory region. In total, we defined 97 independent mutation events in *Lr9*

across the 120 mutants (that is, different nucleotide transitions or identical transitions in the two genetic backgrounds). We identified 19 premature stop codon mutations, 67 nonredundant amino acid substitutions, four splice-site mutations, and one start codon mutation (Fig. 1d, Extended Data Fig. 5 and Supplementary Table 1). We further validated the identity of *Lr9* by virus-induced gene silencing (VIGS). Silencing of *WTK6-vWA* increased leaf rust susceptibility (Fig. 1e). Together, the mutant analysis, genetic mapping and silencing confirm that *WTK6-vWA* is *Lr9*.

The 67 nonredundant amino acid substitution mutations enabled us to establish a detailed map of critical residues required for WTK6-vWA function. One amino acid substitution mapped to the N-terminal loop, 48 in kinase 1, 13 in kinase 2 and 5 in the C-terminal vWA/Vwaint domain. Protein modeling revealed a marked difference in the effects of individual amino acid substitutions between the two kinase domains (Fig. 2). Thirty-one of 48 amino acid substitutions in kinase 1 were predicted to be surface-localized, while most (8 of 13) of the substitutions in kinase 2 affected internal residues (Fig. 2 and Supplementary Table 1). Substitutions at surface-localized amino acids are likely to interfere with protein function, while substitutions of internal amino acids might affect protein structure and stability[24,25]. Seventeen amino acid substitutions in kinase 1 affected conserved subdomains and loops, including ATP-binding sites, compared to only three amino acids for the conserved kinase 2 subdomains (Extended Data Fig. 5)[26]. In conserved subdomain III, kinase 2 carried an aspartic acid (D) instead of the highly conserved glutamic acid (E) found in kinase 1 (Extended Data Fig. 5), indicating that kinase 2 might be a pseudokinase[13]. Most WTK proteins involved in disease resistance in wheat and barley have been suggested to harbor a kinase and a pseudokinase domain[12–18]. Substitution of the subdomain III glutamic acid in the *Arabidopsis* EVR kinase abrogated kinase activity[27]. Thus, we conclude that the two kinase domains may have different roles in WTK6-vWA.

*WTK6-vWA*-like genes represent a very small gene family in bread wheat, with zero to two full-length copies identified in wheat reference assemblies (Supplementary Table 2)[28–33]. All bread wheat *WTK6-vWA* homologs represented *Lr9* orthologs located on chromosome arms 2AL and 2BL. The wheat lines ArinaLrFor and Fielder carried the closest Lr9 orthologs on chromosome 2A, with 90.1% amino acid sequence identity to Lr9. We also identified one to eight genes encoding proteins with a single kinase domain followed by a vWA/Vwaint domain (referred to as kinase-vWAs) in individual assemblies of bread wheat cultivars and various grass species (Extended Data Fig. 6 and Supplementary Table 2)[28–42]. Phylogenetic analyses provide evidence that the fusion of kinase and vWA domains occurred at least three times during the evolution of grasses (Extended Data Fig. 6 and Supplementary Figs. 4 and 5). We hypothesize that kinase-vWA proteins have a role in PAMP-triggered immunity and are targets of pathogen virulence effectors. The acquisition of a second kinase domain in WTK6-vWA might have resulted in a new function, whereby kinase 2 and the vWA/Vwaint domain serve as decoy[23,43], while kinase 1 activity is required to initiate a defense response after effector perception. All five amino acid substitutions in the vWA/Vwaint domain were surface-localized, supporting that this domain might be involved in effector recognition (Fig. 2 and Supplementary Table 1).

We used a diagnostic Kompetitive Allele Specific PCR (KASP) marker derived from a polymorphism that distinguished *WTK6-vWA* alleles from Tc*Lr9* and *Ae. triuncialis* accession TA10438 (Supplementary Note 2) to screen a diversity panel comprising 59 *Ae. umbellulata* and 205 *Ae. triuncialis* accessions (Supplementary Fig. 6 and Supplementary Table 3). No *Ae. triuncialis* accession carried *Lr9*, and only one *Ae. umbellulata* accession (TA1851) was positive for *Lr9*. We generated contig-level assemblies of Tc*Lr9* and the putative *Ae. umbellulata Lr9* donor TA1851 using PacBio circular consensus sequencing (CCS)[44] to determine the genomic sequence of *Lr9*, including surrounding regions (Table 2). We assembled a 5.66-Mb and 14.99-Mb

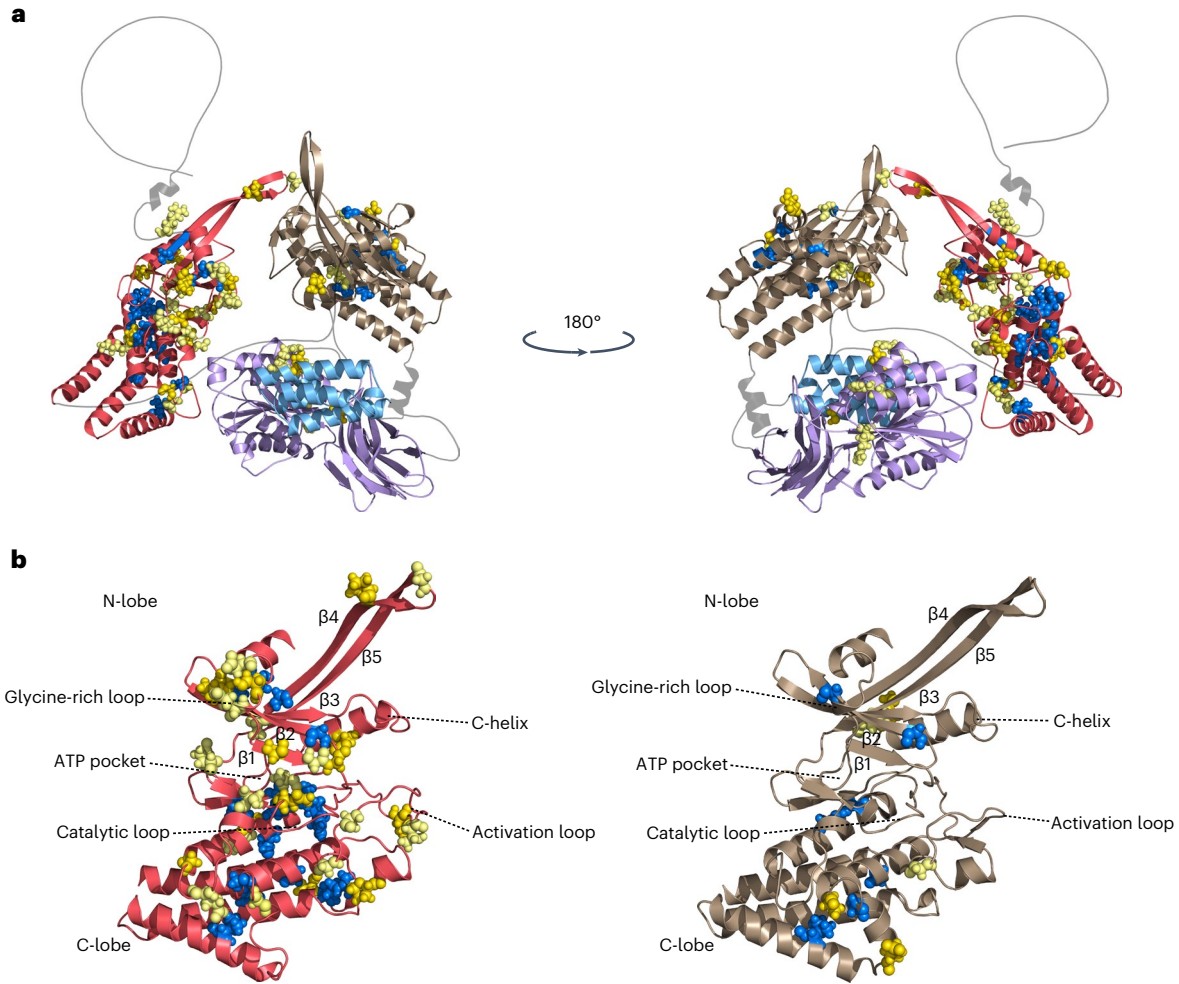

**Fig. 2 | Protein structure prediction and mutation analysis of WTK6-vWA.**
**a**, The protein structure of WTK6-vWA as predicted by AlphaFold v2.0. **b**, Kinase 1 and kinase 2 as predicted by AlphaFold v2.0, indicating the basic secondary functional features. The colors of the domains are identical to Fig. 1d. Salmon red–kinase 1, light brown–kinase 2, purple–vWA domain and light blue–Vwaint domain. Blue spheres indicate internal amino acid substitutions resulting from the EMS mutagenesis. Light and dark yellow spheres represent surface-localized amino acid substitutions.

**Table 2 | Statistics of the Tc*Lr9* and TA1851 assemblies**

| Genomic feature | Tc*Lr9* | TA1851 |
|---|---|---|
| Length of HiFi assembly | 14.49 Gb | 4.25 Gb |
| Number of contigs | 7,686 | 1,379 |
| Contig N50 | 4.52 Mb | 11.15 Mb |
| Contig N90 | 1.2 Mb | 3.18 Mb |
| Largest contig | 30.59 Mb | 57.09 Mb |
| GC content | 46.13% | 47.11% |

*Lr9*-containing contig in Tc*Lr9* and TA1851, respectively (Fig. 3a). The *Lr9* gene sequence spanned 10.63 kb and had 14 exons (Extended Data Fig. 7a). *Lr9* transcript levels slightly increased after inoculation with an avirulent *Pt* isolate (Extended Data Fig. 7b). The *Lr9* gene sequence was identical between Tc*Lr9* and TA1851 and the two contigs showed 99.99% sequence identity over 5.58 Mb, confirming that TA1851 is the *Lr9* donor (Fig. 3a). The few polymorphisms mainly represented short insertion/deletions and are likely considered sequencing errors. The Tc*Lr9* contig had two large deletions of 24.8 and 46.0 kb compared to TA1851, most likely caused by the irradiation treatment (Fig. 3a).

Mapping rates for TA5605, Thatcher and *Ae. triuncialis* TA10438 Illumina reads confirmed that TA5605 carries the same introgression as Tc*Lr9*, while the mapping rates for Thatcher and TA10438 reads were low (Fig. 3b), thus further supporting that *Lr9* and *Lr58* come from the same original introgression.

To reconstruct the history of the *Lr9* translocation, we first estimated the size of the *Lr9* translocation to be ~35–50 Mb by genomic in situ hybridization (GISH; Fig. 3c). We then mapped TA1851 *k*-mers to the Tc*Lr9* assembly, which identified 13 Tc*Lr9* contigs with high *k*-mer mapping rates (Fig. 3d and Supplementary Table 4). The 13 Tc*Lr9* contigs were 0.28–5.66 Mb in size and matched five contigs in TA1851. We ordered and oriented the five TA1851 contigs based on a published *Ae. umbellulata* genetic map[45] (Supplementary Table 5). One Tc*Lr9* contig spanned a 585-bp gap between two neighboring TA1851 contigs, which left four gaps of unknown size. Most of the gaps between the Tc*Lr9* contigs were small (<4 kb) based on the corresponding TA1851 sequence, suggesting that we covered most of the *Lr9* translocation. Tc*Lr9* contig Tc*Lr9*ptg000262l marked the terminal end of the translocation, with *Lr9* locating 6.05 Mb proximal to the end. We identified the proximal breakpoint of the *Ae. umbellulata* translocation in Tc*Lr9* contig Tc*Lr9*ptg002383l (Fig. 3d). *k*-mer mapping rates and dot plots revealed the exact breakpoint between bread wheat chromosome 6B and the *Ae. umbellulata* translocation (Fig. 3e and Extended Data

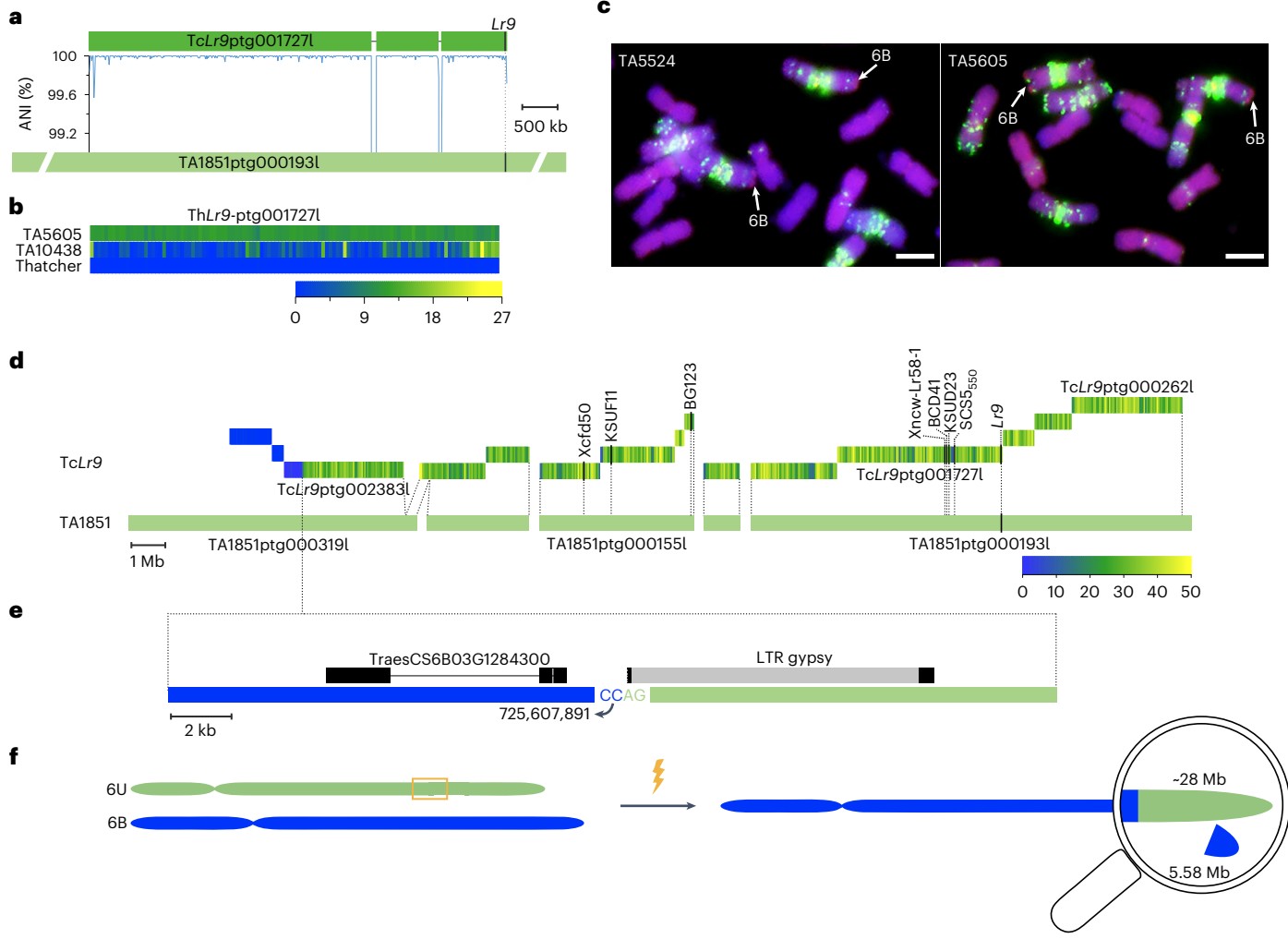

**Fig. 3 | Assembly of the *Lr9* translocation. a**, *Lr9*-carrying contigs identified in the Tc*Lr9* and *Ae. umbellulata* (TA1851) assemblies. The chart in the middle shows the average nucleotide identity (ANI) calculated in nonoverlapping sliding windows of 10 kb. Lines in the Tc*Lr9* contig indicate irradiation-induced deletions. **b**, Heatmap showing Illumina read coverage (in nonoverlapping sliding windows of 50 kb) from TA5605, Thatcher and *Ae. triuncialis* accession TA10438 mapped against the Tc*Lr9* contig. **c**, GISH patterns of partial mitotic metaphase spreads of TA5524 (Transfer) and TA5605 using *Ae. umbellulata* genomic DNA (visualized in red). The *Ae. umbellulata* translocation on chromosome arm 6BL is indicated by arrows. GAA repeats (green signals) were used to identify the homologous chromosome pairs. Experiment was repeated independently twice with similar results. Scale bars = 5 μm. **d**, Thirteen Tc*Lr9* contigs were identified that showed increased coverage of *Ae. umbellulata*-derived *k*-mers. The heatmap

in the Tc*Lr9* contigs represents the *k*-mer coverage in sliding windows of 50 kb. The corresponding TA1851 contigs are indicated below the Tc*Lr9* contigs (ordered and oriented according to an *Ae. umbellulata* genetic map). The positions of published *Lr9* and *Lr58* genetic markers are indicated. **e**, Schematic representation showing the translocation breakpoint between *Ae. umbellulata* (represented in green) and bread wheat chromosome 6BL (represented in blue). The black bars in the gypsy retroelement correspond to the long terminal repeats (LTRs). The nucleotide position 725,607,891 indicates the corresponding position in the Chinese Spring reference assembly (IWGSC v2.1). **f**, Schematic representation of the nonhomologous chromosome translocation between *Ae. umbellulata* chromosome 6U and bread wheat chromosome 6BL. The irradiation led to chromosome breaks and random chromosome fusions. During the translocation, a 5.58-Mb bread wheat chromosome segment was lost.

Fig. 8), which was located in the putative promoter region of an NLR gene in bread wheat, while the breakpoint in *Ae. umbellulata* corresponded to a retrotransposon. These results confirmed that the use of irradiation led to a fusion of nonhomologous chromosome segments. The *Ae. umbellulata* translocation spanned ~28.4 Mb, in agreement with the cytogenetic estimate. Bread wheat line Chinese Spring was used as a recurrent parent to generate Transfer, which allowed us to compare the exact positions of the translocation breakpoints in Tc*Lr9* (carrying the *Lr9* translocation from Transfer) and the Chinese Spring reference genome[28,30]. The translocation breakpoint was at position 725.61 Mb (RefSeq v2.1), which was 5,580,341 bp proximal to the end of Chinese Spring chromosome 6B, indicating that this 5.58-Mb segment including 87 high-confidence genes is missing in *Lr9*-containing wheat lines

(Fig. 3f and Supplementary Table 6). Alien introgressions, including the *Lr9*-containing *Ae. umbellulata* translocation, are often associated with reduced yields[46]. Our results show that such yield decline might be caused both by undesirable alleles on the *Ae. umbellulata* introgression (linkage drag) and by the loss of this 5.58-Mb chromosomal segment.

In summary, we identified an unusual class of kinase fusion proteins involved in disease resistance. Our results indicate that the two kinase domains have diverse functions and we hypothesize that the second kinase and the vWA/Vwaint domain are integrated decoys. A second example of an 'integrated decoy kinase fusion protein' is the protein encoded by the stem rust resistance gene *Sr43*[47]. Sr43 contains a kinase domain fused to two domains of unknown function that might serve as decoys. Together, these two studies highlight the emerging role

of unusual kinase fusion proteins in wheat disease resistance[13–18,48–51], which will form the basis to use this protein class in breeding.

## Online content

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

## Methods

### Plant material

Bread wheat accessions Transfer (TA5524), WL711, TA5605, *Ae. umbellulata* accession TA1851 and *Ae. triuncialis* accession TA10438 were obtained from the Wheat Genetics Resource Center (WGRC). Tc*Lr9* (Transfer/6*Thatcher) is a near-isogenic line carrying *Lr9* from Transfer in the genetic background of the susceptible wheat line Thatcher. Tc*Lr9* and TA5605 were crossed with leaf rust susceptible wheat cultivar Avocet S to develop segregating F2 populations for *Lr9* (136 F2 plants) and *Lr58* (128 F2 plants), respectively. An F2 population derived from a cross between Tc*Lr9* and TA5605 (133 F2 plants) was also generated.

### Leaf rust inoculations

*P. triticina* isolates B9414, 93012, 94015, 95012, 96007 and 96237 are avirulent on *Lr9* (refs. [52],[53]). Isolate MNPSD is virulent on *Lr9* (ref. [54]). Isolates were propagated on seedlings of the susceptible wheat cultivar Thatcher. Freshly collected urediniospores were used for inoculation experiments. Inoculations were performed using a high-pressure air sprayer or a settling tower. For the spray inoculation, urediniospores were suspended in FC-43 oil (3M Fluorinert FC-43) and sprayed onto plants using a glass sprayer connected to a high-pressure air pipe. For the settling tower inoculation, plants were grown for 12 d. Then, the second leaf was removed and the first leaf was pinned to the soil with the adaxial side facing upward. A mixture of 10 mg urediniospores and 300 mg lycopodium powder (Sigma-Aldrich, 19108) was blown over the plants in a settling tower[55]. Inoculated plants were placed in an inoculation box equipped with a humidifier overnight and transferred to a walk-in Conviron growth room (16 h day/8 h night, 18 °C/16 °C). Symptoms were evaluated 12 d after inoculation. Infected leaves were scanned using an Epson Perfection V600 Photo scanner.

### EMS mutagenesis and mutant screening

Grains of wheat lines Tc*Lr9* and TA5605 were evenly divided and soaked in two 1-l flasks with 200 ml water at 4 °C for 16 h. Grains were washed three times in distilled water. Then, 200 ml of 0.45% and 0.55% EMS solutions (Sigma-Aldrich, M0880) were added and grains were incubated on a shaker at 80 rpm at room temperature for 16 h. Then, grains were washed three times for 45 min each, and then transferred into mesh bags and put under running tap water for 30 min. After the washing, grains were put into 55 × 28 cm black plastic trays with blotting paper at the bottom, which were later covered with plastic lids and kept at 4 °C for 36 h. The treated grains were planted into 18-well trays filled with Stender soil, 12 to 14 grains per well. Plants were grown in a greenhouse under speed-breeding conditions (22 h day/2 h night, 21 °C/18 °C)[56]. We performed three EMS experiments for TA5605 (with approximately 2,000, 4,000 and 5,000 grains per experiment), and one for Tc*Lr9* (~2,000 grains). Spikes from the first and second batches of TA5605 M1 plants were bulk collected. Single spikes were collected from the third batch of TA5605 M1 plants and from Tc*Lr9* M1 plants. For the mutant screening, M2 grains from a single spike were planted in one well of 18-well or 24-well trays and grown in a greenhouse (22 h day/2 h night, 21 °C/18 °C) or growth room (16 h day/8 h night, 20 °C) for 3–4 weeks. Batches of six trays were spray-inoculated with 30 mg fresh urediniospores of *Pt* isolate B9414 suspended in 20 ml FC-43 oil, and then kept in an inoculation box overnight. Plants were transferred and grown in a walk-in Conviron growth room for 10–14 d. Susceptible plants were transferred into single pots. In total, 919 and ~7,400 M2 families were screened for Tc*Lr9* and TA5605, respectively.

### Iso-seq library preparation and sequencing

Total RNA from infected Tc*Lr9* and TA5605 seedling leaves (48, 72, 96 and 120 h after inoculation with the avirulent *Pt* isolate B9414) was extracted using the Maxwell RSC Plant RNA Kit (Promega, AS1500). To prepare the Iso-seq libraries, 300 ng of total RNA was used as input for complementary DNA (cDNA) synthesis. Each sample was first barcoded (Supplementary Table 7) and then subjected to cDNA amplification using 12 cycles. Purified cDNAs were pooled in equal molarity and then subjected to library preparation using the SMRTbell Express Template Prep Kit 2.0 (Pacific Biosciences, 100-938-900) following the Iso-seq protocol. Libraries were prepared for sequencing by annealing primer v4 with Sequel II Binding Kit 2.1 and the Internal Control Kit 1.0 (Pacific Biosciences, 101-843-000). Two SMRT Cells 8M (Pacific Biosciences, 101-389-001), one for Tc*Lr9* and one for TA5605, were sequenced on the PacBio Sequel II system using the Sequencing Kit 2.0 (Pacific Biosciences, 101-820-200).

### RNA-seq of susceptible mutants

Total RNA from the *Pt*-inoculated fourth to sixth leaves of ten susceptible Tc*Lr9* M2 mutants and ten TA5605 M3 mutants was extracted using the Maxwell RSC Plant RNA Kit (Promega, AS1500). RNA-seq was performed as a service at Novogene. In brief, mRNA samples were enriched using oligo(dT) beads and then used to prepare strand-specific cDNA libraries. Sequencing was done on Illumina Novaseq 6000, which produced an average of 233.2 and 262.7 million paired-end reads (2 × 150 bp) for Tc*Lr9* and TA5605 mutants, respectively.

### MutIsoSeq

The principle of MutIsoSeq is to use full-length transcripts produced with PacBio Iso-seq as a reference, map RNA-seq reads of mutants to the reference, and identify transcripts that carry multiple independent EMS-type point mutations (Fig. 1c). The workflow of MutIsoSeq is illustrated in Supplementary Fig. 1. The raw data from PacBio Sequel II were processed following the Iso-seq pipeline (v.3; https://github.com/PacificBiosciences/IsoSeq) for the generation of CCS reads and full-length, nonconcatemer (FLNC) reads. FLNC reads were merged and clustered into 'clustered.hq.fasta.gz' files containing 300,066 and 283,330 high-quality transcripts for Tc*Lr9* and TA5605, respectively. Transposable elements were masked by Repeatmasker (v4.0.7; http://www.repeatmasker.org) using the TREP database[57] (v.2019) as an external library, which resulted in 6.29% and 6.08% masked sequences, respectively. Then, SeqKit v2.2.0 (ref. [58]) was used to sort all the transcripts from longest to shortest, and the fasta files generated during this step were used as references. We used BBMap (v.38.96)[59] (parameters−ambiguous=best; subfilter=1; trimreaddescriptions=t nodisk) to map the Illumina reads from the susceptible mutants to the references. To exclude nonspecific alignments as much as possible, we used the parameter 'subfilter=1', which removes alignments with more than one substitution. Iso-seq captures different isoforms originating from the same gene. If a gene had multiple isoforms, Illumina reads would be randomly assigned to different isoforms by default, which would reduce the average coverage per isoform. We thus used the parameter 'ambiguous=best', which assigns reads to the first best site. Because transcripts were ordered by size, this step resulted in the mapping of most reads to the longest isoform of a gene. The resulting SAM files were sorted and indexed by SAMtools (v1.6)[60]. We followed the MutChromSeq pipeline for the remaining steps (https://github.com/steuernb/MutChromSeq)[61]. Briefly, sorted SAM files were converted into the mpileup format using SAMtools (v1.6)[60]. Individual pileup files were subsequently converted into XML format using Pileup2XML.jar (parameter, -a 0.01 -c 5). Reports were given by MutChromSeq.jar using the parameters -n 10 -c 10 -a 0.01 -z 2 (Tc*Lr9*-M13 and TA5605-Spt1 were used with the parameter '-w' because we did not use the wild-type (WT) Tc*Lr9* and TA5605 RNA-seq data) (Supplementary Fig. 2). For both *Lr9* and *Lr58*, only one candidate transcript was identified that had EMS-type G/C-to-A/T transitions in all 20 sequenced mutants. We performed a BLAST search with the identified candidate transcripts against the high-quality transcripts using TBtools (v1.09873)[62], which identified 72 and 61 transcripts for *Lr9* and *Lr58*, respectively (Supplementary Fig. 3a,b). Only transcripts that encoded a full-length protein spanning all the detected mutants were retained, resulting in

16 transcripts each for *Lr9* and *Lr58*. Twelve and 11 of the retained *Lr9* and *Lr58* transcripts encoded the same 1,167 amino acid protein, respectively, and they mainly differed in the length of the 5′ and 3′ untranslated regions (Supplementary Fig. 3c). The transcripts encoding this protein version accounted for the majority of the total transcripts (68.15% and 73.26% coverage, which were calculated using the coverage number of transcripts, representing the number of molecules sequenced in the PacBio SMRT cells) (Supplementary Fig. 3a,b). We also found four *Lr9* transcripts (representing 2.54% coverage) and four *Lr58* transcripts (2.18% coverage) that encoded full-length proteins lacking 12 to 24 amino acids at the N-terminus, and one *Lr58* transcript (0.41% coverage) encoding a full-length protein with 28 extra amino acids. The transcripts encoding the 1,167 amino acid protein were considered the main protein version because of the high transcript abundance.

## Sanger sequencing of EMS mutants

We performed a BLAST search with the *WTK6-vWA* CDS against the genome sequence of bread wheat cultivars[29,31] and found full-length matches on chromosome arms 2AL and 2BL of wheat cultivars ArinaLrFor and Fielder. The corresponding genomic sequences were downloaded and aligned to *WTK6-vWA* full-length transcript/46588 of Tc*Lr9* using Geneious Prime (v.2020.2.4). Primers were designed from the UTRs and the CDS of *WTK6-vWA* and intron sequence of ArinaLrFor and Fielder (Supplementary Table 8). PCR amplifications were done in 20 μl reactions containing 1× GoTaq Green Master Mix (Promega, M7122), ~100 ng genomic DNA and 100 nM primers. Primer pairs K1-F/K1-R, K2-F2/K2-R1 and vWA-F2/vWA-R1 were used to amplify 1,949-bp, 2,313-bp and 1,050-bp fragments from the DNA samples of mutants. Amplicons were Sanger-sequenced using primers K1-F/K1-MR/K1-R, K2-F2/K2-MF1/K2-MR/K2-R1 and vWA-F2/vWA-R1, respectively. For the mutants from the last batch of mutant screening (TA5605-Spt124 to Spt186), primer pair K1-F/K1-R was used to amplify the 1,949-bp fragment, and nested PCR[63] primer pairs K2-F1/vWA-R1 (for the first round PCR) and K2-F2/vWA-R2 (for the second round PCR) were used to amplify a second 3,740-bp fragment from the mutants, which were subsequently Sanger-sequenced using primers K1-exon1-R/K1-MF1/K1-MF2/K1-R and K2-F2/K2-F3/K2-F4/K2-R1/vWA-F1/vWA-R2. Sanger sequencing files were aligned to the *WTK6-vWA* CDS and genomic sequence to find polymorphisms using Geneious Prime (v.2020.2.4). Primer sequences are provided in Supplementary Table 8. We chose a conservative approach and only considered mutants as independent if they contained nucleotide transitions at different positions in the *Lr9* gene or at the same position but in two different genetic backgrounds (TA5605 and Tc*Lr9)*, which resulted in 97 independent mutation events.

## Virus-induced gene silencing

To develop specific VIGS probes, we first performed a BLAST search with the *WTK6-vWA* CDS against various wheat genomes[29]. *WTK6-vWA* sequence stretches with little homology in the wheat assemblies were selected as targets (Fig. 1d). Primers for the two targets and the phytoene desaturase genes (PDS)[64] were used to amplify a 223-bp (VIGS1), 226-bp (VIGS2) and 400-bp (PDS) fragment from the TA5605 cDNA, respectively (Supplementary Table 8). Fragments were cloned into the BSMV-γ (barley stripe mosaic virus (BSMV)) vector[64], resulting in constructs BSMV-γVIGS1, BSMV-γVIGS2 and BSMV-γPDS. These constructs together with BSMV-γ (empty vector), BSMV-γGFP[32], BSMV-α and BSMV-β were transformed into *Agrobacterium tumefaciens* strain GV3101. Agrobacteria were cultured overnight at 28 °C in lysogeny broth with appropriate antibiotics. Cells were collected by centrifugation, then re-suspended and adjusted to $OD_{600}$ = 0.7. Equal volumes of BSMV-α and BSMV-β were mixed with BSMV-γVIGS1, BSMV-γVIGS2, BSMV-γGFP, BSMV-γ and BSMV-γPDS, respectively. After 3 h incubation at 28 °C, cultures were infiltrated into *Nicotiana benthamiana* leaves. Infiltrated leaves were collected 5 d after infiltration and homogenized with PBS buffer (Gibco, 10010023) containing 1% Celite (Thermo

Fisher Scientific, 68855-54-9). The buffer containing viral particles was rub inoculated on seedlings of Tc*Lr9* and TA5605 at two-leaf stage. After 2 weeks, when viral symptoms were clearly visible, plants were spray-inoculated with *P. triticina* isolate B9414 as described above. Disease symptoms were evaluated and recorded 12 d after inoculation.

## Protein structure prediction

To predict the 3D structure of WTK6-vWA, we used the open-source code of AlphaFold v2.0 (ref. [65]). The input was the amino acid sequence of WTK6-vWA. The output consisted of five PDB format text files containing the predicted structure exactly as predicted by the model ('unrelaxed models'), which were later refined by an Amber relaxation procedure ('relaxed models') to provide the relaxed predicted structure reordering by model confidence based on the Local Distance Difference test (lDDT) score ('ranked models'). We used one of the ranked models with a reported lDDT score of 0.7507 of very good accuracy (lDDT > 0.6 is considered reasonable models and lDDT > 0.8 is considered great models, accuracy-wise). The positions of 67 nonredundant amino acid substitutions were visualized as spheres using PyMOL (v.2.3.0). Substituted residues that did not exhibit any visual surface on the 3D structure model were characterized as internal residues.

## Cytogenetic analysis

The procedures for mitotic chromosome preparation, FISH and GISH, were adapted from refs. [66,67]. A 3,740-bp fragment was amplified from Tc*Lr9*, TA5605 and TA10438 using nested PCR[63] primer pairs K2-F1/vWA-R1 (for the first round PCR) and K2-F2/vWA-R2 (for the second round PCR) and Phusion High-Fidelity DNA Polymerase (New England Biolabs, M0530), which was subsequently Sanger-sequenced using primers K2-F2/K2-MR/K2-MF2/K2-R2/K2-R1/LastIntron-F/vWA-F3/vWA-R2 (Supplementary Table 8). The 3,740-bp fragment from Tc*Lr9* was cloned into pJET1.2/blunt (Thermo Fisher Scientific, K1231) and used in FISH as a probe. Probe DNA was labeled with either digoxigenin-11-dUTP or biotin-16-dUTP following the manufacturer's instructions (Roche, CA). Unlabeled total genomic wheat DNA was used as a blocker in GISH experiments. After post-hybridization washes, the probe hybridization signals were detected with Alexafluor 488 streptavidin (Invitrogen) for biotin-labeled probes, and rhodamine-conjugated anti-digoxigenin (Roche) for dig-labeled probes. Images were captured using a Zeiss Axioplan 2 microscope (Carl Zeiss Microscopy) mounted with a cooled CCD camera CoolSNAP HQ2 (Photometrics) and operated with AxioVision 4.8 software. The final contrast of the images was processed using Adobe Photoshop CS5.

## Identification of *WTK6-vWA* homologs

*WTK6-vWA* homologous genes were retrieved by BLAST search against whole-genome assemblies of 12 bread wheat cultivars, durum wheat, wild emmer wheat and six *Aegilops* species at WheatOmics 1.0 (http://202.194.139.32/blast/blast.html)[29–36,68]; two rye cultivars and one hexaploid oat (OT3098) at GrainGenes (https://wheat.pw.usda.gov/blast/)[37,38], *Thinopyrum intermedium*, *Brachypodium distachyon*, *Dichanthelium oligosanthes* and *Eleusine coracana* at Phytozome (https://phytozome-next.jgi.doe.gov)[39,40]; *Eragrostis tef* and *Eragrostis curvula* at GoGe (https://genomevolution.org/coge/)[41,42] and *Ae. umbellulata* TA1851. Corresponding sequences were downloaded. Nonannotated sequences were manually annotated by MUSCLE alignment in Geneious Prime (v.2020.2.4). Translated protein sequences were used as input in the 'NCBI Conserved Domain Search' (https://www.ncbi.nlm.nih.gov/Structure/cdd/wrpsb.cgi) for protein domain annotation.

## Phylogenetic analyses

In total, 186 putative kinase and pseudokinase domains of tandem kinase proteins, tandem kinase-vWAs, and kinase-vWAs from different grass species were used[13–17]. The vWA domains of tandem kinase-vWA and kinase-vWA proteins from Supplementary Table 2, and

vWA-containing proteins retrieved from proteomes of barley Morex v3 (https://wheat.pw.usda.gov/blast/), *Aegilops speltoides* AEG-9674-1 (http://202.194.139.32/blast/blast.html) and rice[69] were used for the phylogenetic analyses. Conserved kinase and vWA domains were identified using NCBI Conserved Domain Search (https://www.ncbi.nlm.nih.gov/Structure/cdd/wrpsb.cgi). Phylogenetic trees (neighbor-joining tree) for kinase and vWA domains were computed with Clustal Omega (https://www.ebi.ac.uk/Tools/msa/clustalo/) and drawn with iTOL (https://itol.embl.de/).

### KASP assay

Five microliters reaction containing 2.5 µl of KASP Master Mix (Low ROX KBS-1016-016), 0.07 µl of assay mix (KASP-Lr9-F/KASP-TA10438-F/KASP-R; Supplementary Table 8) and 2.5 µl (50 ng) of DNA was used for the *Lr9* KASP assay. PCR cycling was performed using an ABI QuantStudio 6 Flex Real-Time PCR machine as follows: preread at 30 °C for 60 s and hold the stage at 94 °C for 15 min, followed by ten touchdown cycles (94 °C for 20 s; touchdown at 61 °C, decreasing by 0.6 °C per cycle for 60 s) and 29 additional cycles (94 °C for 20 s; 55 °C for 60 s). Plates were read at 30 °C for endpoint fluorescence measurement.

### HiFi library preparation, sequencing and assembly of Tc*Lr9* and TA1851

High molecular weight (HMW) DNA was extracted from young seedlings of Tc*Lr9* and TA1851 using a modified Qiagen Genomic DNA extraction protocol (https://doi.org/10.17504/protocols.io.bafmibk6)[70]. DNA was sheared to the appropriate size range (15–20 kb) using Megaruptor 3 (Diagenode) for the construction of PacBio HiFi sequencing libraries. Library preparation was done with the Express Template Prep Kit 2.0 (100-938-900 + Enzyme Clean up 2.0 (101-932-600)), and size was selected with a PippinHT System (Sage Science, HTP0001). Final SMRTbell QC was assessed with Qubit dsDNA High Sensitivity (Thermo Fisher Scientific, Q33230) and FEMTO Pulse (Agilent Technologies, P-0003-0817). Sequencing was performed on PacBio Sequel II/IIe systems in CCS mode. The 169.0 Gb of Tc*Lr9* PacBio HiFi reads and 130.9 Gb of TA1851 PacBio HiFi reads were assembled using hifiasm (v.0.16.1)[71] with default parameters. The genome assemblies were evaluated using QUAST 5.0.2 (ref. 72) and BBMap (v.38.96)[59].

### Re-sequencing of TA5605, Thatcher and TA10438

Genomic DNA was extracted from leaf samples of bread wheat accessions TA5605, Thatcher and *Ae. triuncialis* accession TA10438 using a CTAB DNA extraction method. Library preparation and sequencing were done as a service by Novogene. In brief, sequencing libraries were generated using the NEBNext Ultra II DNA Library Prep Kit and sequenced on an Illumina NovaSeq 6000 system. Illumina sequencing resulted in about 564.3 Gb, 542.5 Gb and 561.8 Gb of 2 × 150 bp paired-end reads for TA5605, Thatcher and TA10438, respectively, corresponding to about 36-fold, 35-fold and 53-fold genome coverage for TA5605, Thatcher and TA10438, respectively.

### Assembly of the *Lr9* translocation

To calculate the average nucleotide identity (ANI) between contigs Tc*Lr9*ptg001727l (Tc*Lr9* contig carrying *Lr9*) and TA1851ptg000193l (TA1851 contig carrying *Lr9*), we aligned them using Minimap2 (v.2.21; parameter '-ax asm5')[73]. InDels and single-nucleotide polymorphisms (SNPs) were manually evaluated in nonoverlapping sliding windows of 10 kb along the two contigs. The assemblies of Tc*Lr9* and TA1851 were merged and used as a reference for the Illumina read mapping of TA5605, Thatcher and TA10438. Illumina reads of TA5605, Thatcher and TA10438 were aligned against the combined PacBio assembly using BBMap (v.38.96; parameters—ambiguous=random; trimreaddescriptions=t; pairedonly=t; perfectmode=t nodisk). From the output SAM file, the read coverage for the contig Tc*Lr9*ptg001727l has been calculated in nonoverlapping sliding windows of 50 kb using

bedtools (v2.30.0)[74] and mosdepth (v. 0.3.3)[75]. The heatmap representing the read coverage was generated with TBtools (v1.09873)[62]. We used a *k*-mer-based approach to reconstruct the *Lr9* translocation. Briefly, we counted the canonical 51-mers from the 130.9 Gb TA1851 raw HiFi reads using Jellyfish (v. 2.2.10)[76], then filtered the 51-mers with at least four occurrences. The dump file containing the 51-mers was converted into fasta format using AWK, which generated 2,726,203,170 *k*-mers. Then, the 51-mers were mapped against the Tc*Lr9* assembly using BWA-MEM (v.0.7.17) allowing only perfect matches (parameter, -k 51 -T 51), generating a bam file of *k*-mer mapping, which was imported to Geneious Prime (v.2022.1.1) and sorted by mean coverage from highest to lowest (Supplementary Table 4). Thirteen contigs with average mapping coverage above 20 were considered originating from *Ae. umbellulata* (Supplementary Table 4). We performed a BLAST search using nonrepetitive sequences of 13 Tc*Lr9* contigs against the TA1851 assembly and identified five corresponding contigs in the *Ae. umbellulata* genome. To validate, order and orient the five selected TA1851 contigs, we anchored 3,009 markers (SNP tags) of the *Ae. umbellulata* linkage map[45] to the TA1851 assembly using BBMap (v.38.96) (parameters—ambiguous=random; trimreaddescriptions=t; subfilter=1 nodisk)[59] (Fig. 3d and Supplementary Table 5). *k*-mer mapping revealed an introgression breakpoint in contig Tc*Lr9*ptg002383l. To further investigate the breakpoint, we extracted 10 kb sequences upstream and downstream of the breakpoint from TA1851ptg000319l, Tc*Lr9*ptg002383l and the long arm of CS chromosome 6B. Dot plot graphs were generated for 20 kb sequences from TA1851 and CS against Tc*Lr9* using Geneious Prime (v.2020.2.4), which revealed the same breakpoint as with *k*-mer mapping (Extended Data Fig. 8). We amplified the PCR markers SCS5$_{550}$ (ref. 77) and Xncw-Lr58-1 (ref. 78) from Tc*Lr9* and TA5605 and sequenced the PCR products using Sanger sequencing. Both markers showed the same sequence from Tc*Lr9* and TA5605, and they were anchored to contigs Tc*Lr9*ptg001727l and TA1851ptg000193l, respectively. The *Lr58* linked markers BCD410, KSUD23, KUSF11, BG123 and Xcfd50 (ref. 3) were also anchored to the Tc*Lr9* contigs using published probe or SSR sequences (Fig. 3d).

### Real-time quantitative PCR

Leaves of Tc*Lr9* were collected 0, 24, 48, 72, 96, 120, 144 and 192 h after inoculation with the avirulent *Pt* isolate B9414. Total RNA was extracted using the Maxwell RSC Plant RNA Kit. First-strand cDNA was synthesized using the iScript cDNA Synthesis Kit (Bio-Rad). RT-qPCR was performed using the primer pairs KKV-qPCR-F/KKV-qPCR-R and Ta-GAPDH-qPCR-F/Ta-GAPDH-qPCR-R (Supplementary Table 8) on a QuantStudio 6 Flex Real-Time PCR system (Applied Biosystems) using PowerUp SYBR Green Master Mix (Thermo Fisher Scientific, A25742). The wheat glyceraldehyde 3-phosphate dehydrogenase gene (GAPDH) was used as the endogenous control. The $2^{-\Delta\Delta CT}$ method was used to normalize and calibrate transcript values relative to the endogenous Ta-GAPDH control[79].

### Reporting summary

Further information on research design is available in the Nature Portfolio Reporting Summary linked to this article.

## Data availability

Data supporting the findings of this work are available within the paper and its Supplementary Information. The raw Iso-seq and RNA-seq data used for MutIsoSeq, the PacBio CCS reads used for de novo whole-genome assemblies, the Illumina raw reads of TA5605, TA10438, Thatcher and sorted TA5605 chromosome 2B were deposited in the European Nucleotide Archive (ENA) under study number PRJEB53839. The *Lr9* genomic and mRNA sequences were deposited in NCBI Genbank under accession numbers ON872164 and ON872165. The genome assemblies of Tc*Lr9* and *Aegilops umbellulata* accession TA1851, the

assembly of sorted TA5605 chromosome 2B, the clustered Iso-seq transcripts of Tc*Lr9* and TA5605 and the CDS and genomic sequence of *Lr9* are available on the DRYAD database under https://doi.org/10.5061/dryad.gxd2547pw.

## Code availability

All software used in this study are publicly available as described in the 'Methods' section and the Reporting Summary.

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

## Acknowledgements

We are grateful to L. Zou (KAUST) for technical assistance, Y. Zhou (KAUST) and H.I. Ahmed (KAUST) for discussions on bioinformatics analyses and the KAUST Bioscience Core Lab for sequencing support. We thank Z. Dubská, R. Šperková and J. Weiserová (Institute of Experimental Botany) for the preparation of chromosome samples for flow cytometry, M. Said and P. Cápál (Institute of Experimental Botany) for chromosome 2B sorting and J. Raupp from the Wheat Genetics Resource Center for providing germplasm. We thank J. A. Kolmer from the USDA-ARS Cereal Disease Laboratory, St. Paul, MN, for providing *P. triticina* isolate MNPSD. We thank P. Lu from IGDB, CAS, for providing TKP sequences for phylogenetic analysis. This publication is based upon work supported by the King Abdullah University of Science and Technology (KAUST) Office of Sponsored Research (OSR) under Award OSR-CRG2018-3768. D.K. was supported by WGRC/IUCRC and NSF (grant 1822162). I.M. received support from the Marie Curie Fellowship grant 'AEGILWHEAT' (H2020-MSCA-IF-2016-746253) and from the Hungarian National Research, Development and Innovation Office (K135057). M.K., K.H. and J.D. received support from the ERDF project 'Plants as a tool for sustainable global development' (CZ.02.1.01/0.0/0.0/16_019/0000827). Computational resources were supplied by the project 'e-Infrastruktura CZ' (e-INFRA LM2018140) provided within the program Projects of Large Research, Development and Innovations Infrastructures.

## Author contributions

Y.W. and S.G.K. designed the overall concept of the research. Y.W. performed molecular experiments and rust inoculations. Y.W. and N.A. generated mapping populations. Y.W., M.A. and E.C. performed assemblies and analyzed genomic data. D.K. and J.P. supplied germplasm and performed fluorescence in situ hybridization and GISH experiments. S.G. and L.J. performed protein modeling. M.K., I.M. and J.D. performed chromosome flow sorting. K.H. sequenced flow-sorted chromosome 2B. Y.W. and S.G.K. wrote the initial version of the manuscript. All authors contributed to subsequent versions and have read and approved the manuscript.

## Competing interests

All authors declare no competing interests.

## Additional information

**Extended data** is available for this paper at https://doi.org/10.1038/s41588-023-01401-2.

**Correspondence and requests for materials** should be addressed to Simon G. Krattinger.

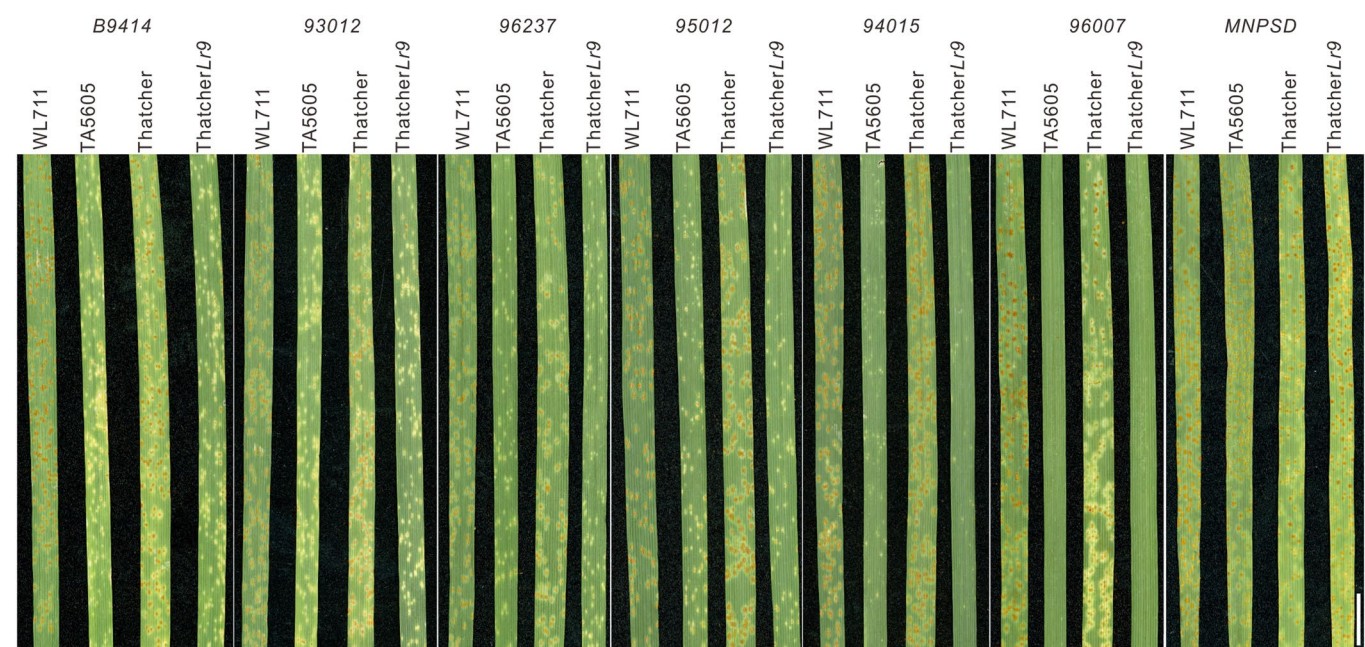

**Extended Data Fig. 1 | *Lr9* and *Lr58* confer resistance to multiple *Puccinia triticina* (*Pt*) isolates.** Seedling inoculations on the *Lr58*-carrying wheat line TA5605 (near isogenic line in the genetic background of bread wheat line WL711) and the *Lr9*-containing bread wheat line Thatcher*Lr9*. Images were taken 12 days after inoculation with seven different *Pt* isolates. Isolates B9414, 93012, 96237, 95012, 94015, and 96007 are avirulent on *Lr9*. *Pt* isolate MNPSD is virulent on *Lr9*. Scale bar = 1 cm.

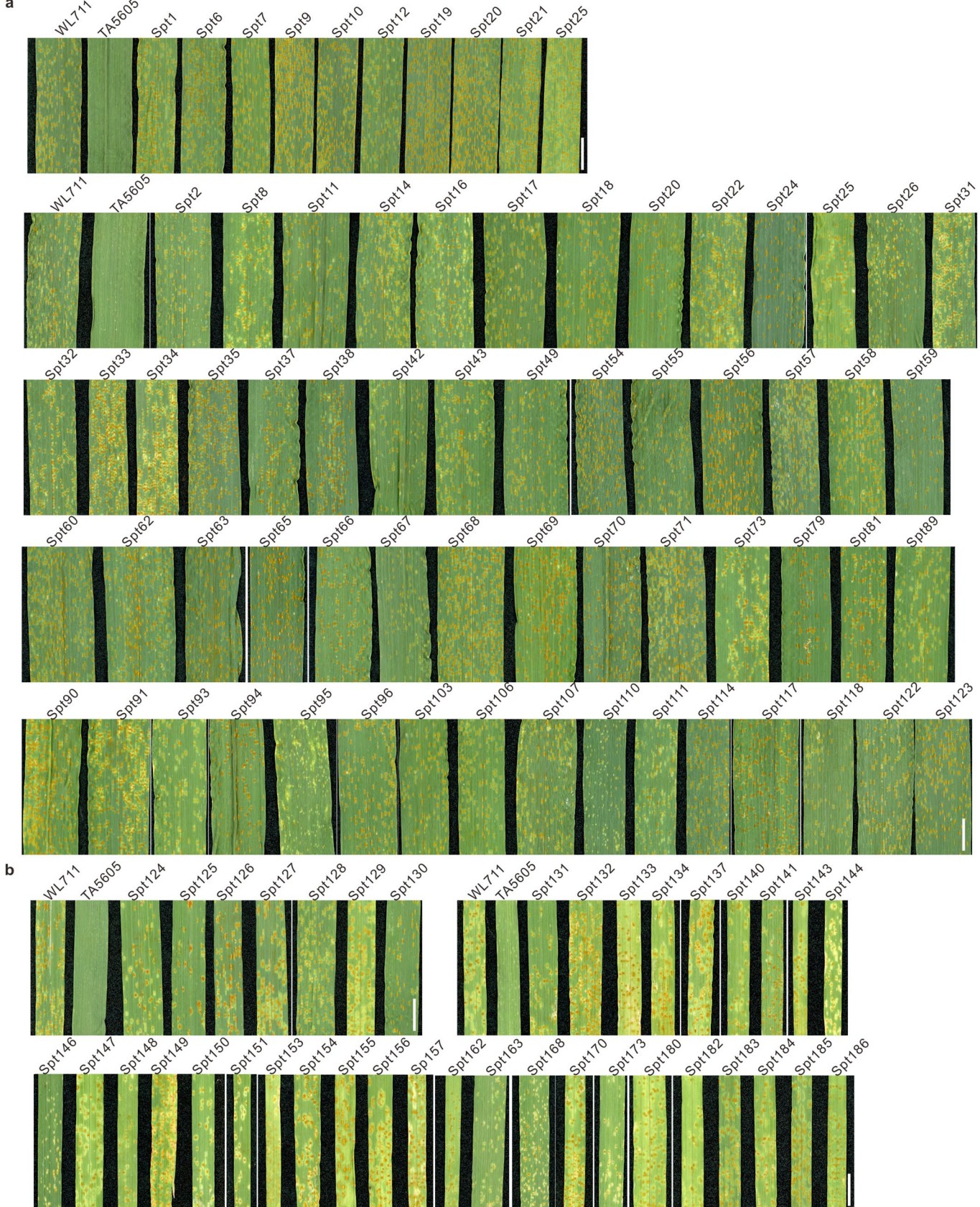

**Extended Data Fig. 2 | Representative images showing leaf rust symptoms on susceptible TA5605 EMS mutants.** TA5605 M3 mutants (**a**) and M2 mutants (**b**) were inoculated with *P. triticina* isolate B9414. Shown are leaves 12 days after inoculation. Spt = Suceptible to *Puccinia triticina*. Scale bar = 1 cm.

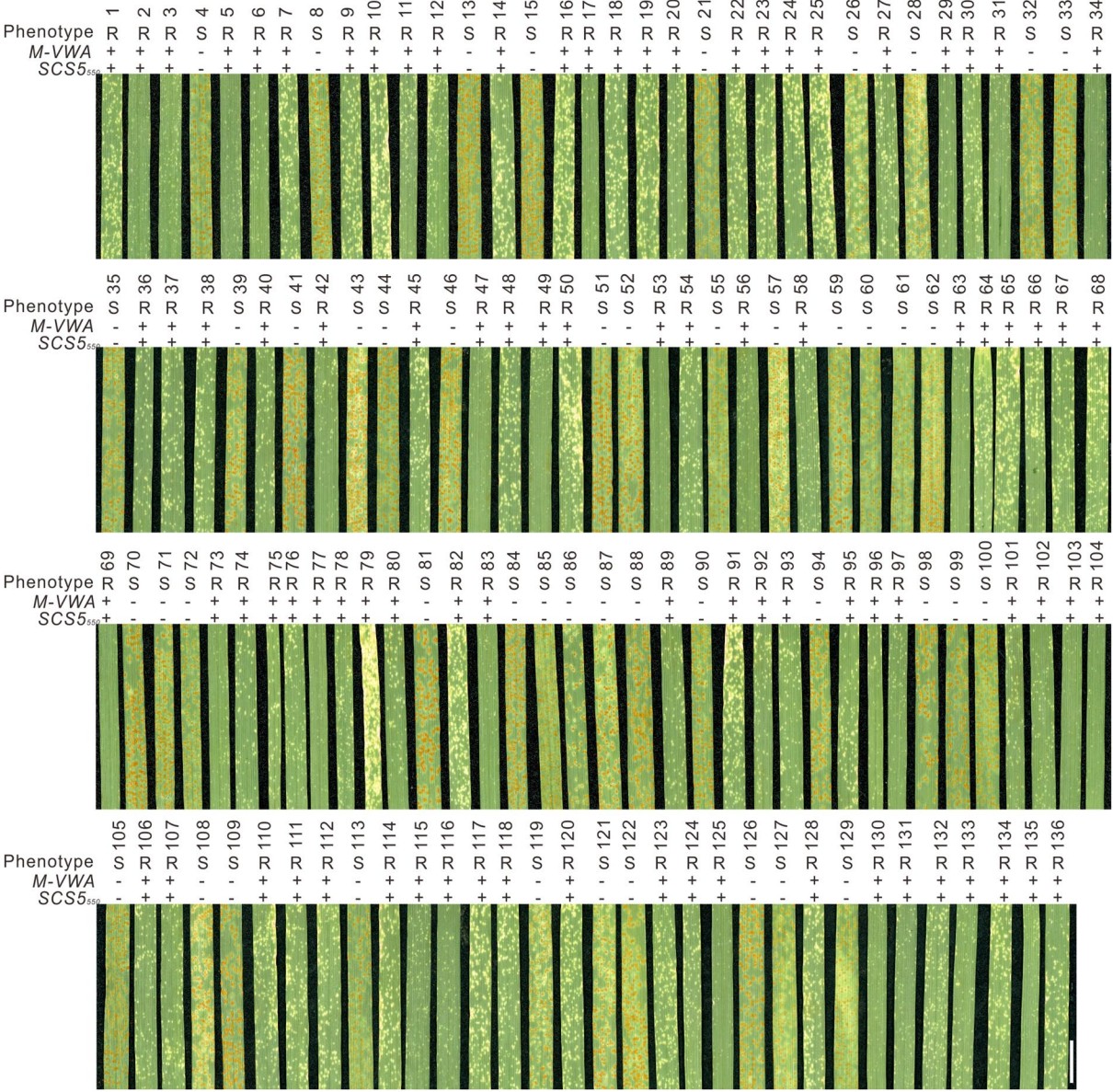

**Extended Data Fig. 3 | WTK6-vWA co-segregated with Lr9-mediated disease resistance and a published Lr9 marker.** Shown are representatives images of an F2 population derived from a cross between ThatcherLr9 and Avocet S inoculated with the avirulent *P. triticina* isolate B9414. M-VWA represents a PCR marker developed from the *WTK6-vWA* coding sequence. SCS5$_{550}$ represents a PCR marker reported to be linked to *Lr9*. Markers were genotyped as present (+) or absent (-). R = resistant, S = susceptible. Images were taken 12 days after inoculation. Scale bar = 1 cm.

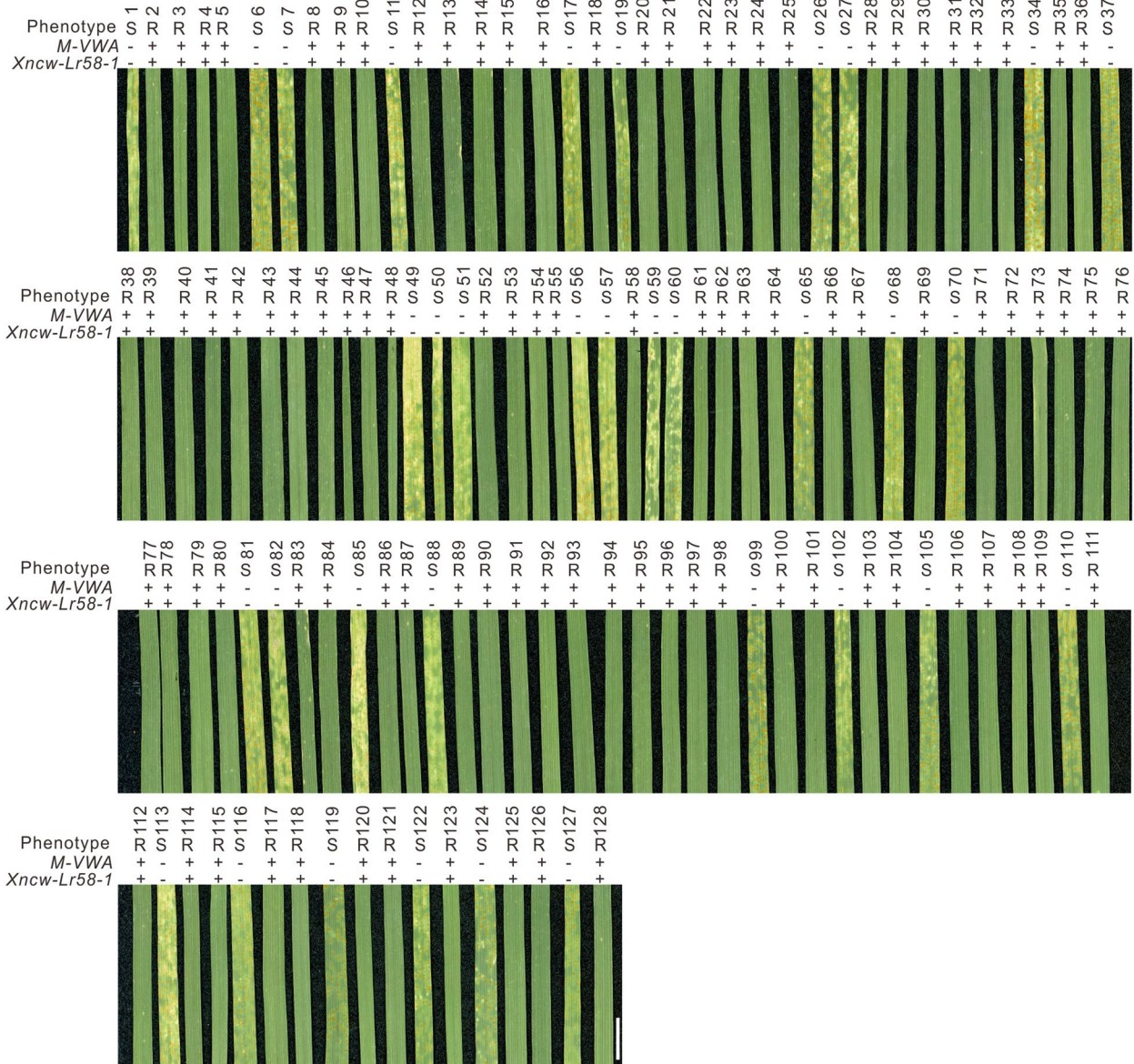

**Extended Data Fig. 4 | *WTK6-vWA* co-segregated with *Lr58*-mediated disease resistance and a published *Lr58* marker.** Shown are representative images of an F2 population derived from a cross between TA5605 and Avocet S inoculated with the avirulent *P. triticina* isolate 96007. M-VWA represents a PCR marker developed from the *WTK6-vWA* coding sequence. Xncw-Lr58-1 represents a PCR marker reported to be linked with *Lr58*. Markers were genotyped as present (+) or absent (-). R = resistant, S = susceptible. Images were taken 12 days after inoculation. Scale bar = 1 cm.

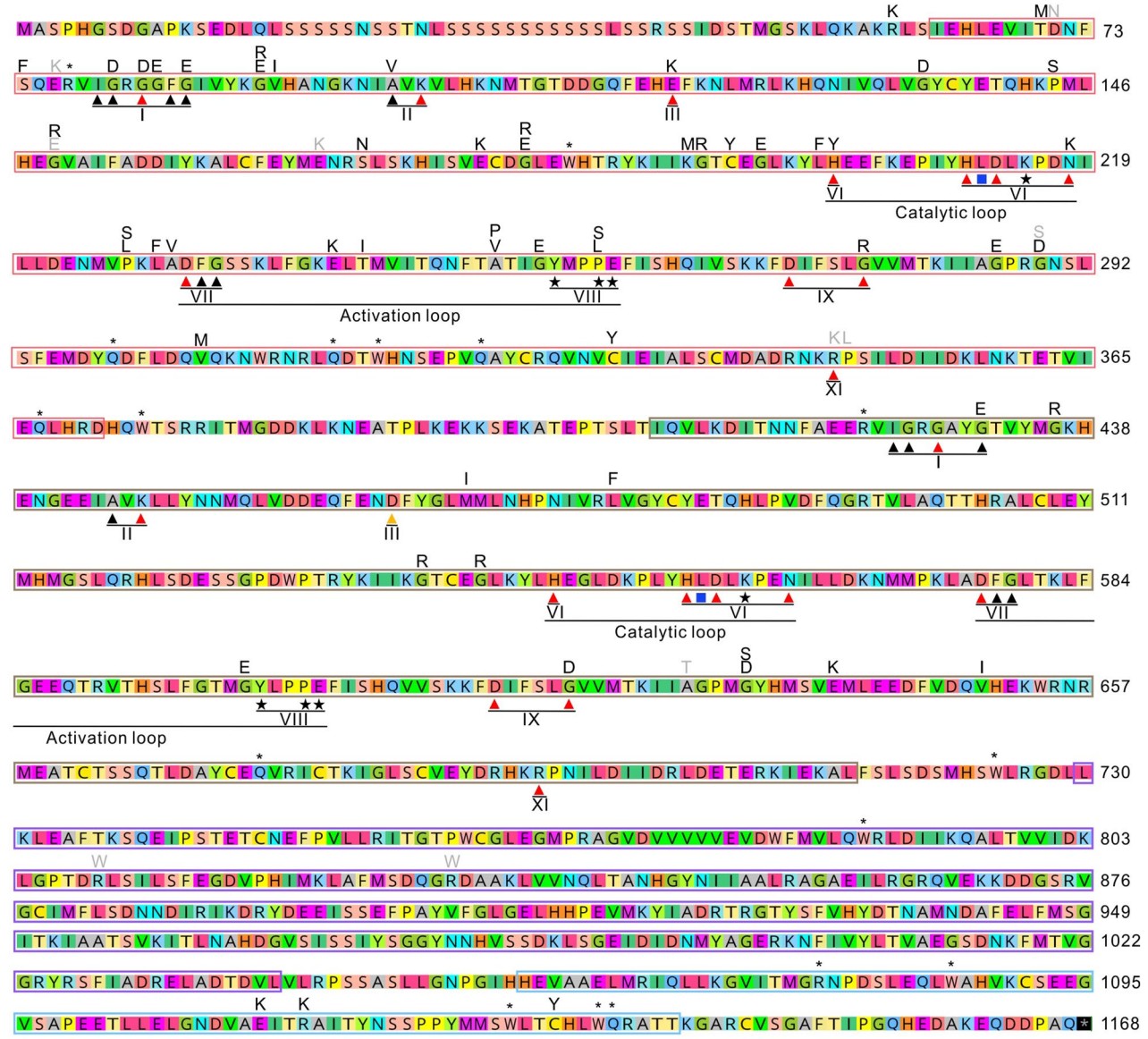

**Extended Data Fig. 5 | Non-redundant mutations in WTK6-vWA identified in the EMS mutagenesis.** The kinase 1 domain is surrounded by a red box, kinase 2 by a brown box, the vWA domain by a purple box, and the Vwaint domain by a blue box. Roman numerals represent conserved kinase subdomains. Letters above the Lr9 amino acid sequence represent EMS-induced amino acid substitutions. Letters in gray were found in partially susceptible mutants, indicating a knock-down of protein function. '*' indicates mutations causing a premature stop codon. Black triangles = ATP binding site; red triangles = key conserved residues, black asterisks = putative substrate binding site, blue squares = residue determining RD and non-RD kinases, orange triangle = glutamic acid to aspartic acid substitution in subdomain III of kinase 2.

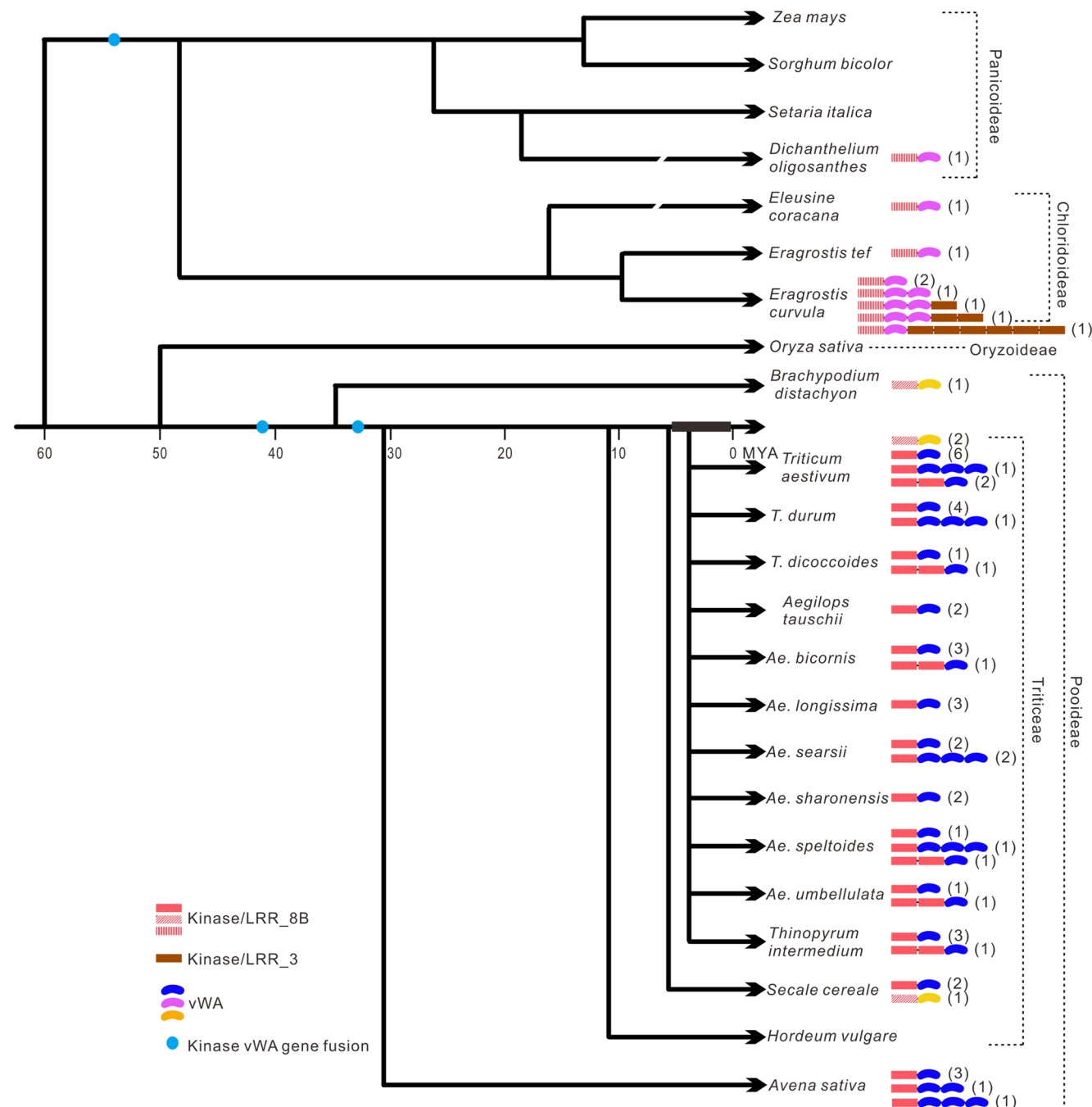

**Extended Data Fig. 6 | Schematic representation showing the origin and distribution of *WTK6-vWA* gene family in Poaceae.** Rectangular boxes in red and brown colors represent kinase domains belonging to the LRR_8B (cysteine rich kinases) and LRR_3 (leucine-rich-repeat receptor kinase subfamily 3) families. Bent ovals in blue, purple and orange represent vWA domains. Blue dots indicate kinase vWA gene fusion events. Numbers in brackets represent the maximum number of genes found in a given accession of each species.

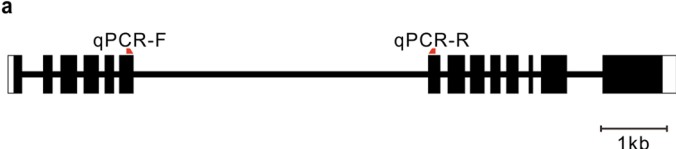

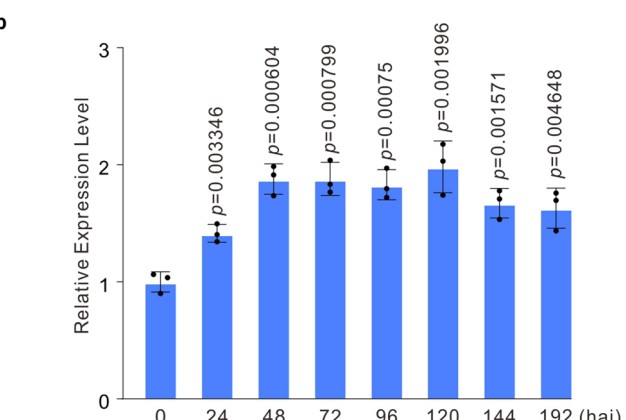

**Extended Data Fig. 7 | Gene structure and RT-qPCR analysis of *WTK6-vWA*.**
**a**, *WTK6-vWA* gene structure. Black boxes = exons; lines = introns; white boxes = 5′UTR and 3′UTR, red arrows indicate the positions of primers used for RT-qPCR. **b**, Relative expression of *WTK6-vWA* in Thatcher*Lr9* after inoculation with the avirulent *P. triticina* isolate B9414. hai = hours after inoculation. Error bars represent standard deviations of three biological replicates. Statistical analysis was done using a two-tailed t-test.

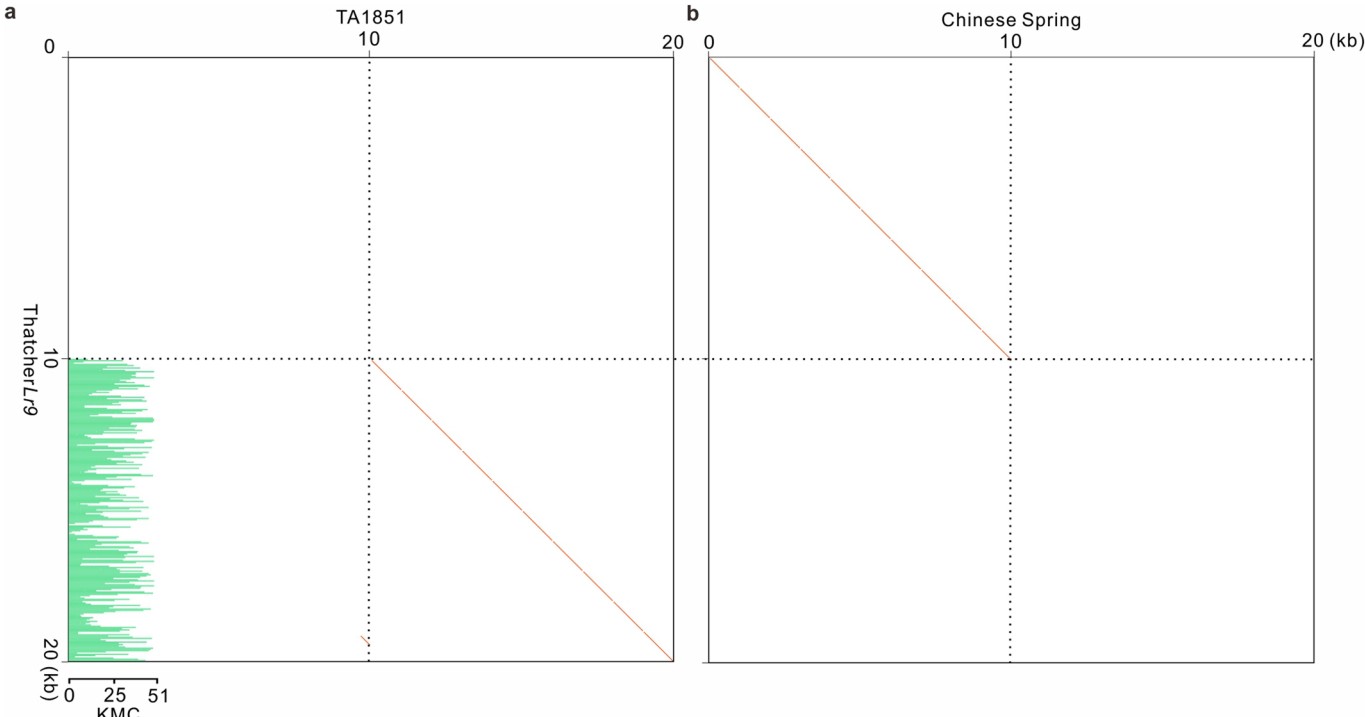

**Extended Data Fig. 8 | k-mer mapping rates and dot plots identified the exact breakpoint between the *Ae. umbellulata* translocation and bread wheat chromosome 6B. a**, Dotplot showing 20 kb sequence alignment between Thatcher*Lr9* and TA1851 around the translocation breakpoint. The green bar chart indicates mean *k*-mer mapping coverage (KMC) of TA1851 *k*-mers acrross the 20 kb region in Thatcher*Lr9*. **b**, Dot plot showing a 20 kb sequence alignment between Thatcher*Lr9* and Chinese Spring around the translocation breakpoint.

# Reporting Summary

## Statistics

For all statistical analyses, confirm that the following items are present in the figure legend, table legend, main text, or Methods section.

| n/a | Confirmed | |
|---|---|---|
| ☐ | ☒ | The exact sample size (*n*) for each experimental group/condition, given as a discrete number and unit of measurement |
| ☒ | ☐ | A statement on whether measurements were taken from distinct samples or whether the same sample was measured repeatedly |
| ☐ | ☒ | The statistical test(s) used AND whether they are one- or two-sided<br>*Only common tests should be described solely by name; describe more complex techniques in the Methods section.* |
| ☒ | ☐ | A description of all covariates tested |
| ☒ | ☐ | A description of any assumptions or corrections, such as tests of normality and adjustment for multiple comparisons |
| ☒ | ☐ | A full description of the statistical parameters including central tendency (e.g. means) or other basic estimates (e.g. regression coefficient) AND variation (e.g. standard deviation) or associated estimates of uncertainty (e.g. confidence intervals) |
| ☐ | ☒ | For null hypothesis testing, the test statistic (e.g. *F*, *t*, *r*) with confidence intervals, effect sizes, degrees of freedom and *P* value noted<br>*Give P values as exact values whenever suitable.* |
| ☒ | ☐ | For Bayesian analysis, information on the choice of priors and Markov chain Monte Carlo settings |
| ☒ | ☐ | For hierarchical and complex designs, identification of the appropriate level for tests and full reporting of outcomes |
| ☒ | ☐ | Estimates of effect sizes (e.g. Cohen's *d*, Pearson's *r*), indicating how they were calculated |

*Our web collection on statistics for biologists contains articles on many of the points above.*

## Software and code

Policy information about availability of computer code

| Data collection | All the leaf images were scanned using an Epson Perfection V600 Photo scanner and corresponding software. |
|---|---|
| Data analysis | MutIsoSeq:<br>IsoSeq v3<br>SeqKit v2.2.0<br>BBMap v.38.96<br>Samtools v1.6<br>Pileup2XML.jar<br>MutChromSeq.jar<br>TBtools v1.09873<br><br>Protein structure prediction and visualization:<br>AlphaFold v2.0<br>PyMOL 2.3.0<br><br>Genome assembly and evaluation:<br>Hifiasm v.0.16.1<br>QUAST 5.0.2<br>BBMap v.38.96<br>Meraculous v2.2.6 |

Lr9 translocation assembly:
Minimap2 v.2.21
Bedtools v2.30.0
Mosdepth v0.3.3
TBtools v1.09873
Jellyfish v. 2.2.10
BWA mem v.0.7.17
Geneious Prime Version 2020.2.4 and Version 2022.1.1
BBMap v.38.96

Sequence alignment analysis:
Geneious Prime Version 2020.2.4

Phylogenetic analysis:
Clustal Omega (https://www.ebi.ac.uk/Tools/msa/clustalo/)
iTOL (https://itol.embl.de/)

Gene projection:
Liftoff v1.6.3

Figure generation/ Drawing:
CorelDRAW 2018
BioRender (https://www.biorender.com/)

For manuscripts utilizing custom algorithms or software that are central to the research but not yet described in published literature, software must be made available to editors and reviewers. We strongly encourage code deposition in a community repository (e.g. GitHub). See the Nature Portfolio guidelines for submitting code & software for further information.

# Data

Policy information about availability of data

All manuscripts must include a data availability statement. This statement should provide the following information, where applicable:
- Accession codes, unique identifiers, or web links for publicly available datasets
- A description of any restrictions on data availability
- For clinical datasets or third party data, please ensure that the statement adheres to our policy

Data supporting the findings of this work are available within the paper and its Supplementary Information.

The raw Iso-seq and RNA-seq data used for MutIsoSeq, the PacBio CCS reads used for de novo whole-genome assemblies, the Illumina raw reads of TA5605, TA10438, Thatcher and sorted TA5605 chromosome 2B were deposited in the European Nucleotide Archive (ENA) under study number PRJEB53839.

The Lr9 genomic and mRNA sequences were deposited in NCBI Genbank under accession numbers ON872164 and ON872165.

The genome assemblies of ThatcherLr9 and Aegilops umbellulata accessionTA1851, the assembly of sorted TA5605 chromosome 2B, the clustered Iso-seq transcripts of ThatcherLr9 and TA5605 and the CDS and genomic sequence of Lr9 are available on the DRYAD database under https://doi.org/doi:10.5061/dryad.gxd2547pw.

The following public databases were used in the study:

whole-genome assemblies of 12 bread wheat cultivars, durum wheat, wild emmer wheat, and six Aegilops species
WheatOmics 1.0 (http://202.194.139.32/blast/blast.html)

whole-genome assemblies of Rye (Lo7 and Weining), hexaploid oat (OT3098) and barley (Morex)
GrainGenes (https://wheat.pw.usda.gov/blast/)

whole-genome assemblies of Thinopyrum intermedium, Brachypodium distachyon, Dichanthelium oligosanthes and Eleusine coracana
Phytozome (https://phytozome-next.jgi.doe.gov)

whole-genome assemblies of Eragrostis tef and Eragrostis curvula
GoGe (https://genomevolution.org/coge/)

# Human research participants

Policy information about studies involving human research participants and Sex and Gender in Research.

Reporting on sex and gender — N/A

Population characteristics — N/A

Recruitment — N/A

| Ethics oversight | N/A |

Note that full information on the approval of the study protocol must also be provided in the manuscript.

# Field-specific reporting

Please select the one below that is the best fit for your research. If you are not sure, read the appropriate sections before making your selection.

☒ Life sciences ☐ Behavioural & social sciences ☐ Ecological, evolutionary & environmental sciences

For a reference copy of the document with all sections, see nature.com/documents/nr-reporting-summary-flat.pdf

# Life sciences study design

All studies must disclose on these points even when the disclosure is negative.

| Sample size | Sample sizes are described in the manuscript. Sizes of ThatcherLr9 x Avocet S, TA5605 x Avocet S and ThatcherLr9 x TA5605 F2 populations are 136, 128 and 133 plants, representing 272, 256, and 266 gametes, respectively. The sizes of the mapping populations were based on literature and based on the calculated recombination frequency (the genetic resolution of a mapping population with 250 gametes is 0.4 cM, which is sufficient to demonstrate linkage between Lr9/Lr58 and a molecular marker derived from the candidate gene). Size of ThatcherLr9 mutant population = 919, size of TA5605 mutant population = ~7,400. We screened an extremely large mutant population to obtain a high number of susceptible mutants with mutations in Lr9, which has been used for protein domain functional interpretation. The population size is bigger than most reported population sizes in similar studies. No statistical method was used to predetermine sample size. |

| Data exclusions | No data were excluded. |

| Replication | The 17 ThatcherLr9 mutants were phenotyped twice at the M2 generation and validated twice at M3 generation. The TA5605-Spt1 to Spt123 mutants were phenotyped once at M2 generation, and validated at least once at M3 generation. The TA5605-Spt124 to 186 mutants were phenotyped twice at M2 generation. All segregating F2 populations were phenotyped and genotyped once. The silencing experiment was performed independently three times with at least five biological replicates per experiment, which all gave similar results. The genomic in situ hybridization experiment was was performed independently twice with with similar results. RT-qPCR analysis of WTK6-vWA was applied once with three biological replicates, and standard deviations were calculated based on values of replicates. |

| Randomization | Inoculated plants were allocated randomly among groups. |

| Blinding | Blinding was performed when phenotyping plants (i.e., the genotype of the plant was not known when the phenotypic data were recorded). |

# Reporting for specific materials, systems and methods

We require information from authors about some types of materials, experimental systems and methods used in many studies. Here, indicate whether each material, system or method listed is relevant to your study. If you are not sure if a list item applies to your research, read the appropriate section before selecting a response.

### Materials & experimental systems

| n/a | Involved in the study |
|---|---|
| ☒ | ☐ Antibodies |
| ☒ | ☐ Eukaryotic cell lines |
| ☒ | ☐ Palaeontology and archaeology |
| ☒ | ☐ Animals and other organisms |
| ☒ | ☐ Clinical data |
| ☒ | ☐ Dual use research of concern |

### Methods

| n/a | Involved in the study |
|---|---|
| ☒ | ☐ ChIP-seq |
| ☒ | ☐ Flow cytometry |
| ☒ | ☐ MRI-based neuroimaging |

