## [Peer Review File · Nature Genetics]

Peer Review Information

Manuscript Title: An unusual tandem kinase fusion protein confers leaf rust resistance in wheat

Corresponding author name(s): Professor Simon Krattinger

Reviewer Comments & Decisions:

Decision Letter, initial version:

26th Aug 2022

Dear Professor Krattinger,

Your Letter, "An unusual tandem kinase fusion protein confers leaf rust resistance in wheat" has now been seen by 3 referees. You will see from their comments below that while they find your work of interest, some important points are raised. We are interested in the possibility of publishing your study in Nature Genetics, but would like to consider your response to these concerns in the form of a revised manuscript before we make a final decision on publication.

To guide the scope of the revisions, the editors discuss the referee reports in detail within the team with a view to identifying key priorities that should be addressed in revision. In this case, we think all three referees have provided constructive reviews aimed at strengthening the experiments and improving the presentation, and we particularly ask that you address their technical comments as thoroughly as possible with appropriate revisions. We hope that you will find the prioritized set of referee points to be useful when revising your study.

We therefore invite you to revise your manuscript taking into account all reviewer and editor comments. Please highlight all changes in the manuscript text file. At this stage we will need you to upload a copy of the manuscript in MS Word .docx or similar editable format.

*2) If you have not done so already please begin to revise your manuscript so that it conforms to our Letter format instructions, available [here](http://www.nature.com/ng/authors/article_types/index.html). Refer also to any guidelines provided in this letter.

[redacted]

We hope to receive your revised manuscript within 3 to 6 months. If you cannot send it within this time, please let us know.

Sincerely,
Wei

Wei Li, PhD
Senior Editor
Nature Genetics
New York, NY 10004, USA

www.nature.com/ng

Reviewers' Comments:

Reviewer #1:

Remarks to the Author:

The authors used mutagenesis and transcriptome sequencing to clone the leaf rust resistance gene Lr9, which was introduced into bread wheat from the wild grass species *Aegilops umbellulata*. The authors demonstrated that Lr58 was identical to Lr9 and originated from the same translocation event. The putative 3,504-bp Lr9/Lr58 coding sequence encoded a 1,167-amino acid protein with an N-terminal tandem kinase domain followed by a von Willebrand factor A (vWA) domain and a Vwaint domain in the C-terminus. Genes encoding proteins with a single kinase domain followed by vWA/Vwaint domains were identified in individual assemblies of bread wheat cultivars and various wild relatives, however, the evolutionary history is not clear, and some of the writing needs to be improved. This work sheds light on the emerging role of kinase fusion proteins in wheat disease resistance, expanding the repertoire of disease resistance genes for breeding. The manuscript is well written, and we recommend it for publication after a minor revision.

Comments and Questions

1. Lines 37-44: The authors proposed a reasonable model to explain the function of WTK6-vWA. However, they did not mention other possible models previously proposed for tandem kinases (e.g. Klymiuk et al 2021, *Molecular Plant-Microbe Interactions*, 34(10), pp.1094-1102). The authors propose that the vWA/Vwaint domains are serving as the decoy ("All five amino acid substitutions in the vWA/Vwaint domains were surfaced-localized, indicating that this domain might be crucial for effector recognition"). We recommend to mention the alternative option that kinase 2 is the main target for the pathogen effector. We would also recommend the authors to test the option that kinase 2 is a pseudokinase, as demonstrated for WTK1 by Klymiuk et al 2018 (*Nature Communications* 9, no. 1: 1-12). We would also recommend classifying the kinase domain into one of kinase subfamilies of *Arabidopsis*.
2. Lines 40-42: The authors hypothesize that a duplication of the kinase domain in WTK6-vWA might have resulted in a new function, whereby kinase 2 and the vWA/Vwaint domains serve as decoy, while kinase 1 activity is required to initiate a defense response after effector perception." The authors cite an old model proposed for NLRs, while ignoring recent tandem kinase protein (TKP) publications that proposed this hypothesis to explain the mode of action of TKPs in plant immunity.
3. Line 52: Please provide some information about the multiple Pt isolates, where were they collected? What do we know about their virulence on different Lr resistance genes? Do they represent different Pt races?
4. Lines 55-56: Are the 17 and 104 susceptible mutants derived from independent M2 families? Fig. 1b and Extended Data Fig. 2 are not providing sufficient information.
5. Lines 63-64: This sentence is not clear and the presentation and explanations for Table 1 are not clear enough and not well written for the general audience of *Nature Genetics*. Please rewrite the sentence and elaborate. We believe that Table 1 belongs in the Extended Data, rather than in the main text. Table 1 indicates that 5-6 mutant lines will be sufficient for MutRNAseq. We would recommend to better explain and discuss this idea. For the TA5605 mutants, it is not well explained why the authors did not identify a single transcript in 10 TA5605 mutants. We propose to write the exact numbers, rather than the different results based on different settings of calculation, and add this

information to the methods and the Extended Data parts.

6. Line 79-80s: Please indicate how many individuals of F2 populations were used.

7. A grammar mistake in Extended Data Fig. 5: "a F2 populations" should be "an F2 population".

8. Lines 98-104: The authors explained the circumstances leading to the wrong localization of Lr58 on chromosome 2B. However, the authors did not present their own data analysis that support this scenario. For example, molecular markers like Xncw-Lr58-1 (or more flanking markers) that showed linkage with Lr58 can be used to search for their locations on the wheat reference genomes.

9. Lines 108-112: In total, the authors developed an impressive number of 121 mutants of which 20 were used for RNAseq analysis and 101 for Sanger-sequencing. However, Table S1 is presenting only 96 mutants and in the text the authors mentioned only 91 mutants. (19 premature stop codon mutations, 67 non-redundant amino acid substitutions, four splice-site mutations, and one start codon mutation). We recommend presenting clear information on these mutants and adding the mutants' information from the RNAseq in Table S1 and also adding the stop codon, splice-site and start codon mutants in Table S1. It is important to mention also potential susceptible mutants without any missense mutations in the WTK6-vWA, as those described for WTK3 and WTK5 since they may represent important partner proteins of WTKs.

10. Line 128: The authors wrote "Thus, we conclude that the two kinase domains may play different roles in WTK6-vWA." We assume that they mean that Kinase 1 has the kinase activity and Kinase 2 may play a different role. This statement is supporting the model presented by Klymiuk et al. 2021 in which one of the duplicated TKP kinases is serving as a decoy for the pathogen effector.

11. We recommend to more information on the sequence identity described in Supplement Table 2. We propose to try to do the evolutionary tree on each domain of the WTK6-vWA separately, in order to make a clear evolutionary story.

12. Lines 138-139: The statement "kinase-vWA proteins play a role in basal plant immunity and are targeted by Pt virulence effectors" imply that they serve as part of the PTI plant immune system and that the kinase-vWA proteins may be evolved from receptor-like kinases (RLKs). To better support this hypothesis, we recommend classifying these kinases into the Arabidopsis/rice kinomes to see if they belong to one of the kinase subfamilies that were shown to function as RLKs in plant immunity.

13. Line 133: The authors conclude that WTK6-vWA homologs are very rare in bread wheat. However, this conclusion is based on less than 20 wheat reference genome assemblies that are currently available. Maybe this statement can be backed by developing a functional molecular marker and a screening of a large core collection of bread wheat accessions. A better evolutionary conclusion can be based on the screening of wild emmer wheat natural populations that are exposed to natural selection forces (genome AABB).

14. Lines 47-50: "Wheat cultivars carrying Lr9 were released in the late 1960s. Lr9 is still effective in many wheat-growing areas, although virulent Pt isolates emerged a few years after the release of the first Lr9-carrying wheat cultivars. Lr58 has not been widely deployed in wheat breeding, and Pt races virulent on Lr58 have been reported". The findings of the current study demonstrate that pedigree data are not accurate and not always can be trusted. In order to claim that there are Pt isolates that are virulent on Lr9/Lr58 the authors should get access to these isolates and validate this statement using the same lines used in the current study to clone Lr9/Lr58.

15. Line 182: We recommend providing a list of the 87 high-confidence genes that are missing in the bread wheat introgression lines and a list of the genes from *Ae. umbellulata* introgression.

16. It will be interesting to test the expression patterns of WTK6-vWA during the Pt infection progress.

Reviewers: Tzion Fahima and Yinghui Li

Reviewer #2:

Remarks to the Author:

The manuscript by Wang et al. reported the cloning of wheat leaf rust resistance gene Lr9, the first disease resistance gene transferred into bread wheat from a wild relative species *Aegilops umbellulata* in the 1950s. Long reads sequencing, mutagenesis and transcriptome sequencing approaches were applied to isolate a tandem kinase protein fused with vWA and Vwaint domains (WTK6-vWA) as the candidate of Lr9. Enriched EMS-induced mutations were identified in the candidate gene WTK6-vWA and VIGS of the WTK6-vWA resulted loss of leaf rust resistance. They also found Lr58, originally reported from *Ae. triuncialis*, is actually the same gene Lr9 from *Ae. umbellulata*.

In general, the study is well designed and the isolation of a WTK fused with vWA domain provides novel information of wheat disease resistance gene repository, especially for the new WTK protein family. In addition to the previously identified WTK proteins confer stem rust, stripe rust and powdery mildew resistance in barley and wheat, this is the first WTK protein providing leaf rust resistance in the Triticeae family.

1. L113-114: The authors did not provide the F2 population size of Avocet S and TA5605, the linkage of WTK6-vWA with the Lr9 may be also due to none recombination of the 6U chromatin and wheat chromosome.
2. No stable transformation of the WTK6-vWA in wheat was performed. Please provide wheat transgenic validation result.
3. The authors identified abundant mutations in the WTK6-vWA protein. How these mutations in the two kinases and vWA domains interrupted the leaf rust resistance function?

Reviewer #3:

Remarks to the Author:

The study conducted EMS mutagenesis and a combination of long/short read RNA-seq for identification of two leaf rust resistance genes (Lr9 and Lr58) in wheat. It is intriguing that the candidates of the two resistance genes are the same. Although these two genes were mapped to different chromosomes in previous studies, this study provides evidence that these two genes have the same origin and from the same introgression event. The causal gene contains tandem kinase and other domains. Mutations at the two tandem kinase domains appear to have distinct consequences on protein structure, indicating the divergent roles of the two domains. Through genotyping a collection of lines, a potential donor was identified and then confirmed. The translocation breakpoint was then determined using PacBio assembled contigs from the translocation line and the donor line.

In this study, the sequencing technologies and bioinformatics tools were well applied to help clone the genes and understand genomic translocation events. The results were logically presented. The study provided evidence to support that Lr9 and Lr58 are the same gene. The study also attempted to exclude the possibility that seeds were mixed up at some points after the initial mapping studies, which is critical and, hopefully, has been ensured through communications among wheat geneticists. Although a handful of leaf rust resistance genes have been cloned and the gene does not provide durable resistance, this study represents a significant contribution to the field. Below includes other comments.

MutIso-seq was not described clearly in the main text. To help reads understand the methodology, Iso-seq using PacBio long reads for a parental line is better to mention in the main text.

The position of the sentence "A PCR marker derived from WTK6-vWA showed complete linkage with disease resistance in F2 populations derived from crosses between ThatcherLr9 and the susceptible wheat cultivar Avocet S and TA5605 with Avocet S" is awkward. What is attempted to convey in the context?

"The fusion of a WTK to another domain, however, is unique and highly unusual." Do most WTK containing genes only contain WTK domains? Is this uniqueness in wheat genomes or plants in general?

The translocation segment originated from 6U. In the supplemental Table 5, the genetic map was from 4U. Any reason for this conflict?

Author Rebuttal to Initial comments

Point-by-point responses (line numbers in our responses refer to the word version with track changes in the "all markup" view)

Reviewer #1:

Remarks to the Author:

The authors used mutagenesis and transcriptome sequencing to clone the leaf rust resistance gene Lr9, which was introduced into bread wheat from the wild grass species *Aegilops umbellulate*. The authors demonstrated that Lr58 was identical to Lr9 and originated from the same translocation event. The putative 3,504-bp Lr9/Lr58 coding sequence encoded a 1,167-amino acid protein with an N-terminal tandem kinase domain followed by a von Willebrand factor A (vWA) domain and a Vwaint domain in the C-terminus. Genes encoding proteins with a single kinase domain followed by vWA/Vwaint domains were identified in individual assemblies of bread wheat cultivars and various wild relatives, however, the evolutionary history is not clear, and some of the writing needs to be improved. This work sheds light on the emerging role of kinase fusion proteins in wheat disease resistance, expanding the repertoire of disease resistance genes for breeding. The manuscript is well written, and we recommend it for publication after a minor revision.

Comments and Questions

1. Lines 37-44: The authors proposed a reasonable model to explain the function of WTK6-vWA. However, they did not mention other possible models previously proposed for tandem kinases (e.g. Klymiuk et al 2021, *Molecular Plant-Microbe Interactions*, 34(10), pp.1094-1102). The authors propose that the vWA/Vwaint domains are serving as the decoy ("All five amino acid substitutions in the

vWA/Vwaint domains were surfaced-localized, indicating that this domain might be crucial for effector recognition"). We recommend to mention the alternative option that kinase 2 is the main target for the pathogen effector. We would also recommend the authors to test the option that kinase 2 is a pseudokinase, as demonstrated for WTK1 by Klymiuk et al 2018 (Nature Communications 9, no. 1: 1-12). We would also recommend classifying the kinase domain into one of kinase subfamilies of Arabidopsis.

>Our response: Dear Tzion and Yinghui, thanks a lot for your valuable and constructive comments. We mentioned in the main text that kinase 2 and the vWA/Vwaint domain might serve as the decoy (lines 149-150). It is very likely that a putative effector interacts with both domains simultaneously and we do not think that these are 'alternative options'. We have rephrased and clarified the respective statement as follows and cited the Klymiuk et al. 2021 paper (lines 147-153): 'We hypothesize that kinase-vWA proteins play a role in basal plant immunity and are targets of pathogen virulence effectors. The acquisition of a second kinase domain in WTK6-vWA might have resulted in a new function, whereby kinase 2 and the vWA/Vwaint domains serve as decoy^{23,40}, while kinase 1 activity is required to initiate a defense response after effector perception. All five amino acid substitutions in the vWA/Vwaint domain were surfaced-localized, supporting that this domain might be involved in effector recognition (Fig. 2, Supplementary Table 1).'

Kinase 2 might indeed be a pseudokinase following the criteria of Klymiuk et al. 2018. We mention this in the revised version in lines 130-132 as follows: 'In conserved subdomain III, kinase 2 carried an aspartic acid (D) instead of the highly conserved glutamic acid (E) found in kinase 1 (Extended Data Fig. 7), indicating that kinase 2 might be a pseudokinase.'

The two kinase domains belong to the LRR_8B subfamily, which is the most frequent kinase found in WTKs. We mention this in the main text as follows (lines 77-79): 'The Lr9/Lr58 kinase domains belong to the LRR_8B subfamily (cysteine-rich receptor-like kinases), which is the most frequent kinase subfamily found in WTK proteins²³'.

2. Lines 40-42: The authors hypothesize that a duplication of the kinase domain in WTK6-vWA might have resulted in a new function, whereby kinase 2 and the vWA/Vwaint domains serve as decoy³⁹, while kinase 1 activity is required to initiate a defense response after effector perception." The authors cite an old model proposed for NLRs, while ignoring recent tandem kinase protein (TKP) publications that proposed this hypothesis to explain the mode of action of TKPs in plant immunity.

>Our response: We now cite a newer model (Klymiuk et al. 2021) that specifically focuses on WTK proteins at this position (line 150).

3. Line 52: Please provide some information about the multiple Pt isolates, where were they collected? What do we know about their virulence on different Lr resistance genes? Do they represent different Pt races?

>Our response: We included this information with appropriate references in the methods section (lines 501-502). Isolates B9414, 93012, 94015, 95012, 96007, and 96237 are avirulent on *Lr9*, while the new isolate MNPSD is virulent. The *P. triticina* strains are referred to as 'isolates' in literature.

4. Lines 55-56: Are the 17 and 104 susceptible mutants derived from independent M2 families? Fig. 1b and Extended Data Fig. 2 are not providing sufficient information.

>Our response: We agree that this point was not very clear. The Thatcher*Lr9* mutants were harvested as individual M2 families. The spikes of TA5605 mutants were bulk harvested (indicated in the methods section, line 525). We have thus chosen a conservative approach and only consider mutants as independent if they carried nucleotide transitions at different positions in the *Lr9* gene or at the same position but in the two different genetic backgrounds (Thatcher*Lr9* and TA5605). Following these classification criteria, we identified 97 independent mutation events in *Lr9*. We have clarified this in the main text as follows (lines 113-114): 'In total, we defined 97 independent mutation events in *Lr9* across the 120 mutants (i.e., different nucleotide transitions or identical transitions in the two genetic backgrounds).' We also indicate the probability that the 97 independent mutations are the result of chance alone (lines 116-117): 'The probability that the 97 independent mutations in *Lr9* are the result of chance alone is $3.1 \cdot 10^{-92}$.'

5. Lines 63-64: This sentence is not clear and the presentation and explanations for Table 1 are not clear enough and not well written for the general audience of Nature Genetics. Please rewrite the sentence and elaborate. We believe that Table 1 belongs in the Extended Data, rather than in the main text. Table 1 indicates that 5-6 mutant lines will be sufficient for MutRNAseq. We would recommend to better explain and discuss this idea. For the TA5605 mutants, it is not well explained why the authors did not identify a single transcript in 10 TA5605 mutants. We propose to write the exact numbers, rather than the different results based on different settings of calculation, and add this information to the methods and the Extended Data parts.

>Our response: We agree that the MutIsoSeq pipeline needs to be explained in more detail and that the explanation in table 1 was not clear (see also the comment of reviewer 3). The strict word

limit of a Nature Genetics letter unfortunately does not allow us to significantly expand the main text. We have therefore written a supplementary note (Supplementary Note 1) that provides a step-by-step explanation of the MutIsoSeq pipeline based on our data. The supplementary note also includes a calculation showing that the probability of detecting a false positive becomes very low with >5 independent mutants. We think that table 1 shows an essential result and we would prefer to keep it as a main table. However, we have improved the description of the table to make it clearer. In particular, we clarify why the TA5605 candidate transcript was detected in only 8 mutants. Please note that there was a mistake in the original version of the table that affected the total number of transcripts in Thatcher*Lr9*, which was caused by a compromised file (this mistake has no effects on the results).

6. Line 79-80s: Please indicate how many individuals of F2 populations were used.

>Our response: This information is now included in the main text.

7. A grammar mistake in Extended Data Fig. 5: "a F2 populations" should be "an F2 population".

>Our response: Typo has been corrected.

8. Lines 98-104: The authors explained the circumstances leading to the wrong localization of *Lr58* on chromosome 2B. However, the authors did not present their own data analysis that support this scenario. For example, molecular markers like Xncw-*Lr58*-1 (or more flanking markers) that showed linkage with *Lr58* can be used to search for their locations on the wheat reference genomes.

>Our response: Searching for the *Lr58* flanking markers in the wheat reference genomes might be misleading because the *Lr9* introgression represents a non-homologous translocation. The *Lr9*-containing *Ae. umbellulata* segment is homoeologous to the bread wheat chromosomes 2. The best BLAST hits with the *Lr58* flanking markers would thus be on the group 2 chromosomes of bread wheat, which is exactly what might have led to a wrong conclusion in the *Lr58* publication.

9. Lines 108-112: In total, the authors developed an impressive number of 121 mutants of which 20 were used for RNAseq analysis and 101 for Sanger-sequencing. However, Table S1 is presenting only 96 mutants and in the text the authors mentioned only 91 mutants. (19 premature stop codon mutations, 67 non-redundant amino acid substitutions, four splice-site mutations, and one start codon mutation).

We recommend presenting clear information on these mutants and adding the mutants' information from the RNAseq in Table S1 and also adding the stop codon, splice-site and start codon mutants in Table S1. It is important to mention also potential susceptible mutants without any missense mutations in the WTK6-vWA, as those described for WTK3 and WTK5 since they may represent important partner proteins of WTKs.

>Our response: In the previous version, we grouped mutants with a SNP at the same position in one line. This has been changed and Supplementary Table 1 now shows each mutant separately, including the stop codon and splice-site mutations. Only one mutant did not have a SNP in the *Lr9* gene (mentioned in lines 111-113). We did not identify other susceptible mutants. *Lr9* is thus different from other WTK genes, where half of the susceptible mutants represented second-site mutations.

10. Line 128: The authors wrote "Thus, we conclude that the two kinase domains may play different roles in WTK6-vWA." We assume that they mean that Kinase 1 has the kinase activity and Kinase 2 may play a different role. This statement is supporting the model presented by Klymiuk et al. 2021 in which one of the duplicated TKP kinases is serving as a decoy for the pathogen effector.

>Our response: We agree and the Klymiuk et al. 2021 paper is now cited.

11. We recommend to more information on the sequence identity described in Supplement Table 2. We propose to try to do the evolutionary tree on each domain of the WTK6-vWA separately, in order to make a clear evolutionary story.

>Our response: We have constructed an evolutionary tree based on the WTK6-wWA kinase domains and various other kinase domains. These analyses suggest that kinase-vWA fusions occurred at least three times in grasses. An evolutionary model and the phylogenetic trees are shown in the new Extended Data Fig. 8 and Supplementary Figs. 4, 5. In the main text we included the following statement (lines 145-147): 'Phylogenetic analyses provide evidence that the fusion of kinase and vWA domains occurred at least three times during the evolution of grasses (Extended Data Fig. 8, Supplementary Figs. 4, 5).'

12. Lines 138-139: The statement "kinase-vWA proteins play a role in basal plant immunity and are targeted by Pt virulence effectors" imply that they serve as part of the PTI plant immune system and that the kinase-vWA proteins may be evolved from receptor-like kinases (RLKs). To better support this

hypothesis, we recommend classifying these kinases into the Arabidopsis/rice kinomes to see if they belong to one of the kinase subfamilies that were shown to function as RLKs in plant immunity.

>Our response: We have done the classification and the two kinase domains belong to the LRR_8B subfamily found in cysteine-rich receptor-like kinases. This is now mentioned in the main text as follows (lines 77-79): 'The Lr9/Lr58 kinase domains belong to the LRR_8B subfamily (cysteine-rich receptor-like kinases), which is the most frequent kinase subfamily found in WTK proteins²³'.

13. Line 133: The authors conclude that WTK6-vWA homologs are very rare in bread wheat. However, this conclusion is based on less than 20 wheat reference genome assemblies that are currently available. Maybe this statement can be backed by developing a functional molecular marker and a screening of a large core collection of bread wheat accessions. A better evolutionary conclusion can be based on the screening of wild emmer wheat natural populations that are exposed to natural selection forces (genome AABB).

>Our response: The term 'rare' might have been misleading here. We did not refer to the frequency of *WTK6-vWA* in the wheat gene pool, but to the size of the *WTK6-vWA* gene family (i.e., each bread wheat genome only contains 0-2 *WTK6-vW*-like genes). We have rephrased our statement as follows (lines 138-140). '*WTK6-vWA*-like genes represent a very small gene family in bread wheat, with zero to two full-length copies identified in wheat reference assemblies (Supplementary Table 2)'. Since the *Ae. umbellulata* introgression was done artificially, we do not expect the functional *Lr9* gene to be present in wild emmer.

14. Lines 47-50: "Wheat cultivars carrying Lr9 were released in the late 1960s. Lr9 is still effective in many wheat-growing areas, although virulent Pt isolates emerged a few years after the release of the first Lr9-carrying wheat cultivars. Lr58 has not been widely deployed in wheat breeding, and Pt races virulent on Lr58 have been reported". The findings of the current study demonstrate that pedigree data are not accurate and not always can be trusted. In order to claim that there are Pt isolates that are virulent on Lr9/Lr58 the authors should get access to these isolates and validate this statement using the same lines used in the current study to clone Lr9/Lr58.

>Our response: We have inoculated our *Lr9/Lr58* lines with a virulent *Pt* isolate and they are susceptible. The result with the new isolate is shown in the new version of Extended Data Fig. 1.

15. Line 182: We recommend providing a list of the 87 high-confidence genes that are missing in the bread wheat introgression lines and a list of the genes from *Ae. umbellulata* introgression.

>Our response: We have included the new Supplementary Table 6 that lists the 87 Chinese Spring high-confidence genes that are missing as well as the genes on the *Ae. umbellulata* introgression (gene projection using Chinese Spring genes projected onto the ThatcherLr9 contig assembly).

16. It will be interesting to test the expression patterns of WTK6-vWA during the *Pt* infection progress.

>Our response: We have performed an additional experiment and measured *WTK6-vWA* transcript levels at different time points after inoculation with an avirulent *Pt* isolate. The results are summarized in the new Extended Data Fig. 9b and mentioned in the main text as follows (lines 168-169): '*Lr9* transcript levels slightly increased after inoculation with an avirulent *Pt* isolate (Extended Data Fig. 9b).'

Reviewer #2:

Remarks to the Author: The manuscript by Wang et al. reported the cloning of wheat leaf rust resistance gene *Lr9*, the first disease resistance gene transferred into bread wheat from a wild relative species *Aegilops umbellulata* in the 1950s. Long reads sequencing, mutagenesis and transcriptome sequencing approaches were applied to isolate a tandem kinase protein fused with vWA and Vwaint domains (WTK6-vWA) as the candidate of *Lr9*. Enriched EMS-induced mutations were identified in the candidate gene WTK6-vWA and VIGS of the WTK6-vWA resulted loss of leaf rust resistance. They also found *Lr58*, originally reported from *Ae. triuncialis*, is actually the same gene *Lr9* from *Ae. umbellulata*.

In general, the study is well designed and the isolation of a WTK fused with vWA domain provides novel information of wheat disease resistance gene repository, especially for the new WTK protein family. In addition to the previously identified WTK proteins confer stem rust, stripe rust and powdery mildew resistance in barley and wheat, this is the first WTK protein providing leaf rust resistance in the Triticeae family.

1. L113-114: The authors did not provide the F2 population size of Avocet S and TA5605, the linkage of

WTK6-vWA with the Lr9 may be also due to none recombination of the 6U chromatin and wheat chromosome.

>Our response: We added the sizes of the F2 populations in the main text. The population sizes were previously indicated in the Extended Data Figs. 3 and 4. It is correct that the *Ae. umbellulata* 6U chromatin introgression will most likely not recombine. However, the introgression is very small (28.4 Mb = 0.19% of a bread wheat genome). The only candidate transcript that was identified through the whole-transcriptome MutlsoSeq approach mapped to this small 6U introgression.

2. No stable transformation of the WTK6-vWA in wheat was performed. Please provide wheat transgenic validation result.

>Our response: We are of the strong opinion that the EMS mutagenesis and VIGS provide sufficient proof that *WTK6-vWA* is *Lr9*: 1) There has been a lot of controversy over the correct identity of several 'cloned' disease resistance genes in wheat where the candidate gene validation was based on transgenic complementation, including *Yr10* (Liu et al. 2014, Molecular Plant 7: 1740-1755; Yuan et al. 2018, TAG 131:1253-1262), *Pm21* (Cao et al. 2011, PNAS 108: 7727-7732; Xing et al. 2018 Molecular Plant 11: 874-878; He et al. 2018, Molecular Plant 11: 879-882) and *Fhb1* (summarized in Lagudah and Krattinger 2019, Nature Genetics 51: 1070-1071). These examples show that gene validation based on transgenic complementation can be fraught with dangers of misinterpretation. On the other hand, we are not aware of a single example where a disease resistance gene in wheat that was validated with >5 independent EMS mutants was later proven to be wrong. 2) The validation of candidate genes using multiple independent EMS mutants is well accepted in literature. In the past six years, 11 wheat and barley disease resistance genes have been validated and published based on mutagenesis alone or mutagenesis in combination with VIGS (*Pm2*, *Lr22a*, *Rph1*, *Yr5*, *YrSP*, *Yr7*, *Sm1*, *SuSr-D1*, *Lr14a*, *Yr27*, *Snn3-D1*). 3) The probability to find 97 independent mutations (see our clarification on the number of independent mutants raised by reviewer 1) in the same gene by chance alone is $3.1 \cdot 10^{-92}$. In other words, one would have to look at $\sim 1.3 \cdot 10^{86}$ batches of 97 mutants in order to find a false positive where 97 independent mutations occurred in a single gene by chance. The global annual wheat production, for a comparison, is only around $8.6 \cdot 10^{14}$ plants. We have included a statement about the probability of finding 97 independent mutations in the same gene by chance in the main text as follows (lines 116-117): 'The probability that the 97 independent mutations in *Lr9* are the result of chance alone is $3.1 \cdot 10^{-92}$ '. In the methods section (lines 532-535), we added 'The calculation of the probability to obtain 97 independent mutations by chance alone was based on Sanchez-Martin et al. (2016)⁵⁴, considering a GC mutation frequency of $2.43 \cdot 10^{-5}$, a GC content of 44.1% and a contig length of 10.63 kb (size of *Lr9* gene).'

3. The authors identified abundant mutations in the WTK6-vWA protein. How these mutations in the two kinases and vWA domains interrupted the leaf rust resistance function?

>Our response: All mutations lead to a loss or a strong reduction of the *Lr9*-mediated disease resistance. As described in the main text (lines 121-136), most of the amino acid substitution mutations might affect the function of kinase 1, the protein structure or stability, or the putative effector binding site.

Reviewer #3:

Remarks to the Author:

The study conducted EMS mutagenesis and a combination of long/short read RNA-seq for identification of two leaf rust resistance genes (*Lr9* and *Lr58*) in wheat. It is intriguing that the candidates of the two resistance genes are the same. Although these two genes were mapped to different chromosomes in previous studies, this study provides evidence that these two genes have the same origin and from the same introgression event. The causal gene contains tandem kinase and other domains. Mutations at the two tandem kinase domains appear to have distinct consequences on protein structure, indicating the divergent roles of the two domains. Through genotyping a collection of lines, a potential donor was identified and then confirmed. The translocation breakpoint was then determined using PacBio assembled contigs from the translocation line and the donor line.

In this study, the sequencing technologies and bioinformatics tools were well applied to help clone the genes and understand genomic translocation events. The results were logically presented. The study provided evidence to support that *Lr9* and *Lr58* are the same gene. The study also attempted to exclude the possibility that seeds were mixed up at some points after the initial mapping studies, which is critical and, hopefully, has been ensured through communications among wheat geneticists. Although a handful of leaf rust resistance genes have been cloned and the gene does not provide durable resistance, this study represents a significant contribution to the field. Below includes other comments.

Mutlso-seq was not described clearly in the main text. To help reads understand the methodology, Iso-seq using PacBio long reads for a parental line is better to mention in the main text.

>Our response: We agree that the MutlsoSeq pipeline needs to be explained in more detail (see

also comment of reviewer 1). However, the strict word limit of a Nature Genetics letter does not allow us to expand the main text at this position. We have therefore written a supplementary note (Supplementary Note 1) that provides a step-by-step explanation of the MutlsoSeq pipeline and uses the exact numbers.

The position of the sentence “A PCR marker derived from WTK6-vWA showed complete linkage with disease resistance in F2 populations derived from crosses between ThatcherLr9 and the susceptible wheat cultivar Avocet S and TA5605 with Avocet S” is awkward. What is attempted to convey in the context?

>Our response: The message that we want to convey here is that the only candidate transcript identified through our whole-transcriptome sequencing approach genetically co-segregated with the *Lr9*-mediated disease resistance. This provides additional evidence that we have identified the correct gene. We have clarified this point by modifying the sentence as follows (lines 80-84): ‘A PCR marker derived from *WTK6-vWA* showed complete linkage with disease resistance in F2 populations derived from crosses between Thatcher*Lr9* and the susceptible wheat cultivar Avocet S (136 F2 plants) and TA5605 with Avocet S (128 F2 plants) (Extended Data Figs. 3, 4), confirming that the *WTK6-vWA* candidate identified through whole-transcriptome sequencing co-segregates with *Lr9/Lr58*-mediated disease resistance.’”

“The fusion of a WTK to another domain, however, is unique and highly unusual.” Do most WTK containing genes only contain WTK domains? Is this uniqueness in wheat genomes or plants in general?

>Our response: Correct, all WTK gene involved in disease resistance so far only had two kinase domains. There is no example outside the Triticeae tribe where a WTK gene is involved in disease resistance.

The translocation segment originated from 6U. In the supplemental Table 5, the genetic map was from 4U. Any reason for this conflict?

>Our response: This is not a conflict, but rather an inconsistency in the nomenclature of chromosomes across wheat relatives. Chromosome 6U of *Ae. umbellulata* is highly rearranged compared to bread wheat and shows homology to bread wheat chromosomes 2, 4, 6, and 7. The majority of *Ae. umbellulata* chromosome 6U is homologous to the bread wheat group 4 chromosomes (see Fig. 4 of Said et al. 2021 Front Plant Sci 12:689031). However, Edae et al. (2016) assigned the *Ae. umbellulata* chromosome numbers in the genetic map based on the homology to wheat, which is why their chromosome 4U corresponds to chromosome 6U. We have mentioned this inconsistency in nomenclature in Supplementary Table 5. For the main text,

however, we decided to use the designation 'chromosome 6U' to be consistent with the nomenclature used in the previous *Lr9* literature

Decision Letter, first revision:

31st Jan 2023

Dear Dr. Krattinger,

Thank you for submitting your revised manuscript "An unusual tandem kinase fusion protein confers leaf rust resistance in wheat" (NG-LE60491R). It has now been seen by the original referees and their comments are below. The reviewers find that the paper has improved in revision, and therefore we'll be happy in principle to publish it in Nature Genetics, pending minor revisions to satisfy the referees' final requests and to comply with our editorial and formatting guidelines.

Sincerely,
Wei

Wei Li, PhD
Senior Editor
Nature Genetics
New York, NY 10004, USA
www.nature.com/ng

Reviewer #1 (Remarks to the Author):

The authors have addressed all the concerns raised in our previous review to our full satisfaction. We highly recommend the manuscript now for publication in Nature Genetics.

Reviewer #2 (Remarks to the Author):

The revised manuscript by Wang et al. provides additional information and clarify the unclear interpterion. The scientific quality and writing were significantly improved. I was satisfied with the responses of authors and recommend an acceptance of publication after a few sentences' revision.

The authors write "A PCR marker derived from WTK6-vWA showed complete linkage with disease resistance in F2 populations derived from crosses between ThatcherLr9 and the susceptible wheat cultivar Avocet S (136 F2 plants) and TA5605 with Avocet S (128 F2 plants) (Extended Data Figs. 3, 4), confirming that the WTK6-vWA candidate identified through whole-transcriptome sequencing co-

segregates with Lr9/Lr58-mediated disease resistance.”

The WTK6-vWA marker is co-segregated with the leaf rust resistance in 136 F2 plants of ThatcherLr9 x Avocet S and 128 F2 plants of TA5605 x Avocet S could only indicate the 28.4 Mb introgression fragment is associated the leaf rust resistance, not WTK-vWA itself is the only candidate of Lr9/Lr58. Please rewrite this sentence to avoid misunderstanding.

Reviewer #3 (Remarks to the Author):

My questions/comments were well addressed. In particular, Supplementary Note 1 clarifies the process of MutIsoSeq. Here are two additional suggestions:

1. I suggested not to add “The probability that the 97 independent mutations in Lr9 are the result of chance alone is $3.1 \cdot 10^{-92}$.”. Adding this statement is not necessary and requires some assumption.
2. “Lr9 transcript levels slightly increased after inoculation with an avirulent Pt isolate (Extended Data Fig. 9b).”. No Extended Data Fig. 9b was provided.

Author Rebuttal, first revision:

Reviewer #2:

The revised manuscript by Wang et al. provides additional information and clarify the unclear interpretation. The scientific quality and writing were significantly improved. I was satisfied with the responses of authors and recommend an acceptance of publication after a few sentences’ revision. The authors write “A PCR marker derived from WTK6-vWA showed complete linkage with disease resistance in F2 populations derived from crosses between ThatcherLr9 and the susceptible wheat cultivar Avocet S (136 F2 plants) and TA5605 with Avocet S (128 F2 plants) (Extended Data Figs. 3, 4), confirming that the WTK6-vWA candidate identified through whole-transcriptome sequencing co-segregates with Lr9/Lr58-mediated disease resistance.” The WTK6-vWA marker is co-segregated with the leaf rust resistance in 136 F2 plants of ThatcherLr9 x Avocet S and 128 F2 plants of TA5605 x Avocet S could only indicate the 28.4 Mb introgression fragment is associated the leaf rust resistance, not WTK-vWA itself is the only candidate of Lr9/Lr58. Please rewrite this sentence to avoid misunderstanding.

Our response:

The reviewer comment is valid. We have modified our statement as follows (lines 82-87): ‘A PCR marker derived from WTK6-vWA showed complete linkage with disease resistance in F2 populations derived from crosses between ThatcherLr9 and the susceptible wheat cultivar Avocet S (136 F2 plants) and TA5605 with Avocet S (128 F2 plants) (Extended Data Figs. 3, 4), confirming that WTK6-vWA identified through whole-

transcriptome sequencing co-segregates with the genes mediating leaf rust resistance in ThatcherLr9 and TA5605'

Reviewer #3: My questions/comments were well addressed. In particular, Supplementary Note 1 clarifies the process of MutIsoSeq. Here are two additional suggestions:

1. I suggested not to add "The probability that the 97 independent mutations in Lr9 are the result of chance alone is $3.1 \cdot 10^{-92}$ ". Adding this statement is not necessary and requires some assumption.

Our response:

We have removed the respective sentence.

2. "Lr9 transcript levels slightly increased after inoculation with an avirulent Pt isolate (Extended Data Fig. 9b)". No Extended Data Fig. 9b was provided.

Our response:

We double checked and Extended Data Fig. 9b was provided in the revised manuscript (now Extended Data Fig. 7b).

Final Decision Letter:

18th Apr 2023

Dear Dr. Krattinger,

I am delighted to say that your manuscript "An unusual tandem kinase fusion protein confers leaf rust resistance in wheat" has been accepted for publication in an upcoming issue of Nature Genetics.

Your paper will be published online after we receive your corrections and will appear in print in the next available issue. You can find out your date of online publication by contacting the Nature Press Office (press@nature.com) after sending your e-proof corrections. Now is the time to inform your Public Relations or Press Office about your paper, as they might be interested in promoting its publication. This will allow them time to prepare an accurate and satisfactory press release. Include your manuscript tracking number (NG-LE60491R1) and the name of the journal, which they will need when they contact our Press Office.

Please note that *Nature Genetics* is a Transformative Journal (TJ). Authors may publish their research with us through the traditional subscription access route or make their paper immediately open access through payment of an article-processing charge (APC). Authors will not be required to make a final decision about access to their article until it has been accepted. [Find out more about Transformative Journals](https://www.springernature.com/gp/open-research/transformative-journals)

Authors may need to take specific actions to achieve [compliance with funder and institutional open access mandates](https://www.springernature.com/gp/open-research/funding/policy-compliance-faqs). If your research is supported by a funder that requires immediate open access (e.g. according to [Plan S principles](https://www.springernature.com/gp/open-research/plan-s-compliance)) then you should select the gold OA route, and we will direct you to the compliant route where possible. For authors selecting the subscription publication route, the journal's standard licensing terms will need to be accepted, including [self-archiving-and-license-to-publish](https://www.nature.com/nature-portfolio/editorial-policies/self-archiving-and-license-to-publish). Those licensing terms will supersede any other terms that the author or any third party may assert apply to any version of the manuscript.

Please note that Nature Portfolio offers an immediate open access option only for papers that were first submitted after 1 January, 2021.

If you have not already done so, we invite you to upload the step-by-step protocols used in this manuscript to the Protocols Exchange, part of our on-line web resource, [natureprotocols.com](https://www.nature.com/natureprotocols). If you complete the upload by the time you receive your manuscript proofs, we can insert links in your article that lead directly to the protocol details. Your protocol will be made freely available upon publication of your paper. By participating in [natureprotocols.com](https://www.nature.com/natureprotocols), you are enabling researchers to more readily reproduce or adapt the methodology you use. [Natureprotocols.com](https://www.nature.com/natureprotocols) is fully searchable, providing your protocols and paper with increased utility and visibility. Please submit your protocol to <https://protocolexchange.researchsquare.com/>. After entering your [nature.com](https://www.nature.com) username and password you will need to enter your manuscript number (NG-LE60491R1). Further information can be found at <https://www.nature.com/nature-portfolio/editorial-policies/reporting-standards#protocols>

Sincerely,
Wei

Wei Li, PhD
Senior Editor
Nature Genetics
New York, NY 10004, USA
www.nature.com/ng